

# MODIS Cloud-Gap Filled Snow-Cover Products: Advantages and Uncertainties

Dorothy K. Hall[1,2,3], George A. Riggs[3,2], Nicolo E. DiGirolamo[3,2] and Miguel O. Román[4]

[1]Earth System Science Interdisciplinary Center, University of Maryland, College Park, MD 20740, USA
[2]Cryospheric Sciences Laboratory, NASA / Goddard Space Flight Center, Greenbelt, MD 20771, USA
[3]SSAI, Lanham, MD 20706, USA
[4]Earth from Space Institute / USRA, 7178 Columbia Gateway Dr., Columbia, MD 21046, USA

*Correspondence to*: Dorothy K. Hall (dkhall1@umd.edu)

**Abstract.** MODerate resolution Imaging Spectroradiometer (MODIS) cryosphere products that have been available since the launch of the Terra MODIS in 2000 and the Aqua MODIS in 2002 include snow-cover extent (swath, daily and eight-day composites) and daily snow albedo. These products are used in hydrological modeling and studies of local and regional climate, and are increasingly being used to study regional hydrological and climatological changes over time. Reprocessing of the complete snow-cover data record, from Collection 5 (C5) to Collection 6 (C6) and Collection 6.1 (C6.1), has led to improvements in the MODIS product suite. Suomi National Polar-orbiting Partnership (S-NPP) Visible Infrared Imaging Radiometer Suite (VIIRS) Collection 1 (C1) snow-cover products have been available since 2011, and are currently being reprocessed for Collection 2 (C2). To address the need for a cloud-reduced or cloud-free daily snow product for both MODIS and VIIRS, a new daily cloud-gap filled snow-cover product was developed for MODIS C6.1 and VIIRS C2 processing. MOD10A1F (Terra) and MYD10A1F (Aqua) are daily, 500-m resolution cloud-gap filled (CGF) snow-cover map products from MODIS. VNP10A1F is the 375-m resolution CGF snow map from VIIRS. The CGF maps provide daily cloud-free snow maps, along with cloud-persistence maps showing the age of the snow or non-snow observation in each pixel. Work is ongoing to evaluate and document uncertainties in the MODIS and VIIRS standard daily CGF snow-cover products. Analysis of the MOD/MYD10A1F products for study areas in the western United States shows excellent results in terms of accuracy of snow-cover mapping. When there are frequent clear-sky episodes, MODIS is able to capture enough clear views of the surface to produce accurate snow-cover information and snow maps. Even in the extensively-cloud-covered northeastern United States during winter months, snow maps from MODIS CGF products are useful, though the snow maps are likely to miss some snow, particularly during the spring snowmelt period when snow may fall and melt within a day or two, before the clouds clear from the storm that deposited the snow. Comparisons between the Terra and Aqua CGF snow maps have revealed differences that are related to cloud masking in the two algorithms. We conclude that the MODIS Terra CGF is the more accurate MODIS snow-cover product, and should therefore be the basis of an Environmental Science Data Record that will extend the CGF data record from the Terra MODIS beginning in 2000 through the VIIRS era, at least through the early 2030s.

## 1 Introduction




Regular snow-cover mapping of the Northern Hemisphere from space began in 1966 when the National Oceanic and
Atmospheric Administration (NOAA) began producing weekly snow maps to improve weather forecasting (Matson
and Wiesnet, 1981). A 53-year climate-data record (CDR) of Northern Hemisphere snow-cover extent (SCE), based
on NOAA's snow maps is now available at the Rutgers University Global Snow Lab (Robinson et al., 1993; Estilow
et al., 2015). Since the 1960s, snow-cover mapping from space has become increasingly sophisticated. Not only
has the temporal resolution of snow maps increased from weekly to twice-daily, but the spatial resolution has also
improved over time. Data from multiple satellite platforms and instruments with visible/near-infrared (VNIR) and
short-wave infrared (SWIR) bands are now available to support improved snow mapping and snow/cloud
discrimination as compared to the earliest satellite snow-cover maps when sparse satellite data were available.

Due to increasing global temperatures, especially in more-northerly areas in the Northern Hemisphere, the Rutgers
CDR has been used by researchers to show that SCE has been declining and melt has been occurring earlier (Déry
and Brown, 2007). This shortening of the snow season has many implications such as, for example, in the western
United States (Mote et al., 2005; Stewart, 2009; Hall et al., 2015), with earlier snowmelt contributing to a longer fire
season (Westerling et al., 2006; O'Leary et al., 2018).

Medium-resolution SCE maps are produced daily from multiple satellite sensors such as are on the MODerate-
resolution Imaging Spectroradiometer (MODIS) on both the Terra, launched in 1999, and Aqua, launched in 2002,
and the Visible Infrared Imaging Radiometer Suite (VIIRS) on the Suomi - National Polar Partnership (S-NPP) and
the Joint Polar Satellite System – 1 (JPSS-1) satellites, launched in 2011 and 2017, respectively. These snow maps
are used extensively by modelers and hydrologists to study regional and local SCE and to develop snow-cover
depletion curves for multiple hydrological and climatological applications. Algorithms utilizing data from these
sensors provide global swath-based snow-cover maps at spatial resolutions ranging from 375 m to 1 km under clear
skies. Instruments on the Landsat series of satellites and other higher-resolution sensors, such as from the Sentinel
series, provide still-higher spatial resolution, though lower temporal resolution.

Cloud cover is the single most-important factor affecting the ability to map SCE accurately using VNIR and SWIR
sensors. Clouds often create daily gaps in SCE maps that are generated using data only from VNIR and SWIR
sensors. One way to mitigate the cloud issue is through cloud-gap filling (CGF). In this paper, we describe the
MODIS Terra and Aqua CGF algorithm, data products and uncertainties. In addition to the inherent uncertainties in
the MODIS snow maps, discussed elsewhere (e.g., Hall and Riggs, 2007, and in many other papers), there are
additional uncertainties related to gap filling. We also discuss the development of a moderate-resolution
Environmental Science Data Record (ESDR) of SCE and using MODIS and VIIRS standard snow-cover maps.
JPSS launches containing VIIRS sensors are planned through at least 2031, continuing the SCE record at moderate
spatial resolution.



## 2 Background

The MODIS instruments have been providing daily snow maps at a variety of temporal and spatial resolutions beginning on 24 February 2000 following the 18 December 1999 launch of the Terra spacecraft. A second MODIS was launched on 4 May 2002 on the Aqua spacecraft. The MODIS sensors provide a large suite of land, atmosphere, and ocean products [https://modis.gsfc.nasa.gov], including daily maps of global snow cover and sea ice. The prefix, MOD, refers to a Terra MODIS algorithm or product and MYD refers to an Aqua MODIS algorithm or product. When the discussion in this paper refers to both the Terra and Aqua products it will be designated as such using the M*D nomenclature. Information on the full MODIS standard cryosphere product suite is available elsewhere [https://modis-snow-ice.gsfc.nasa.gov/].

Since the launches of the Terra and Aqua spacecraft, there have been several reprocessings of the entire suite of MODIS Land Data Products [https://modis-land.gsfc.nasa.gov/]. Specifically, reprocessing from Collection 5 (C5) to Collection 6 (C6) and in the near future, Collection 6.1 (C6.1), has led to improvements in the MODIS snow-cover standard data products (Riggs et al., 2017 and 2018).

A great deal of validation has been conducted on the MODIS snow-cover products through the C5 era (e.g., Klein and Barnett, 2003; Parajka and Blöschl, 2006; Hall and Riggs, 2007; Frei and Lee, 2010; Arsenault et al., 2012 and 2014; Parajka et al., 2012; Chelamallu et al., 2013; Dietz et al., 2013), including validation with higher-resolution imagery, such as from Landsat Thematic Mapper, Enhanced Thematic Mapper Plus and Operational Land Imager (TM/ETM+ and OLI) (e.g., see Huang et al., 2011; Crawford, 2015; Coll and Li, 2018). Crawford (2015) found strong spatial and temporal agreement between MODIS Terra snow-cover fraction and Landsat TM/ETM+ derived snow cover, noting that some high-altitude cirrus cloud contamination was observed and transient snow was sometimes difficult for the MODIS algorithm to detect. Though use of higher-resolution data is valuable, use of meteorological-station data for validation (e.g., Brubaker et al., 2005) is the only true validation of the snow-cover products. Comparing extent of snow cover derived from MODIS with snow cover from other satellite products is not true validation because all derived snow-cover products have uncertainties.

A new feature of the MODIS C6 product suite provides the snow decision on each map as a normalized-difference snow index (NDSI) value instead of fractional-snow cover (FSC) (Riggs et al., 2017). This has the important advantage of allowing a user to more-accurately determine FSC in their particular study area by applying an algorithm to derive FSC from the NDSI that is tuned to a specific study area. The C5 FSC algorithm (Salomonson and Appel, 2004) is useful for estimating FSC globally for MODIS Terra data products, but is of more-limited utility for specific and especially well-characterized study areas. That algorithm remains useful globally and can easily be applied to the MODIS C6 and C6.1 and VIIRS C1 and C2 NDSI data.





S-NPP VIIRS C1 SCE products [https://doi.org/10.5067/VIIRS/VNP10.001] are designed to correspond to the
MODIS C6 SCE products (Riggs et al., 2017).  There were many revisions made in the MODIS C6 algorithms that
improved snow-cover detection accuracy and information content of the data products.  Though there are important
differences between the MODIS and VIIRS instruments (e.g., the VIIRS 375 m native resolution compared to
MODIS 500 m), the snow-detection algorithms and data products are designed to be as similar as possible so that
the 19+ year MODIS ESDR of global SCE can be extended into the future with the S-NPP and Joint Polar Satellite
System (JPSS)-1 VIIRS snow products and with products from future JPSS platforms.

**2.1 Methods to reduce or eliminate cloud cover in MODIS-derived snow-cover maps**

The presence of cloud cover prevents daily continuous SCE maps from being produced using VNIR and SWIR
sensors.  To reduce the effects of cloud cover in the MODIS snow-cover maps, many researchers have employed a
variety of different methods.  For example, as part of the early MODIS snow-product suite, eight-day maximum
snow-cover maps (M*D10A2) were designed to provide greatly-reduced cloud cover.  However these maps are
available only once every eight days, the maps frequently retain some cloud cover, and it is difficult to determine on
which days during the eight-day period snow was or was not observed.  In spite of this, the eight-day maximum
snow maps have been useful in numerous research studies, e.g., M*D10A2 has been used successfully to develop
snowmelt-timing maps (O'Leary et al., 2018) and to map snow zones (Hammond et al., 2018), and are still available
in C6.0 and 6.1.

Many other methods have also been developed to reduce or eliminate cloud cover in the MODIS snow-cover
product suite.  Parajka and Blöschl (2008) used a 7-day temporal filter causing a reduction of cloud coverage of
>95%, maintaining an overall accuracy of  >92% when SCE was compared with in-situ data.  Other methods to
reduce cloud cover have also been successful (e.g., see for example, Tong et al., 2009a & b; Coll and Li, 2018).
Gafurov and Bárdossy (2009) developed a cloud-clearing method consisting of six sequential steps that begins with
using Terra and Aqua snow cover maps, ground observations, spatial analysis and finally snow climatology to clear
clouds and generate a cloud-free daily snow-cover map with high accuracy.  Gafurov et al. (2016) developed an
operational daily snow-cover monitoring tool using that same cloud-clearing method with enhancements, with a
mean accuracy of 94% for a case study of the Karadarya River basin in Central Asia.  To fill gaps caused by cloud
cover, use of forward and backward gap-filling methods to eliminate cloud cover have been used successfully with
the MODIS standard snow products and other satellite data.

Foppa and Seiz (2012) developed a temporal forward and backward gap-fill method to create a "cloud-free" daily
snow map from the daily global MOD10C1 data product.  A spatial method that uses the relative position of a cloud-
obscured pixel to the regional snow-line elevation (SNOWL) was developed by Parajka et al. (2010) using MODIS
Terra data to create "cloud-free" snow maps which produced robust snow-cover mapping even in situations of
extensive cloud cover.  A combination of SNOWL and temporal forward and backward gap filling was used by



Hüsler et al. (2014) to create "cloud free" satellite snow cover maps using data from the Advanced Very High
Resolution Radiometer (AVHRR) of the European Alps. Malnes et al. (2016) used a multi-temporal
forward/backward interpolation gap-filling technique to create a cloud-free daily snow map from MOD10A1
products that was then used to detect the first and last snow-free day in a season for northern Norway.

A cubic spline interpolation method has been used with good results by some researchers (Tang et al., 2013 & 2017;
Xu et al., 2017) as a temporal CGF method using MODIS snow-cover products. Some researchers have developed
CGF techniques that combined Terra and Aqua, time interpolation, spatial interpolation and probability estimation,
e.g. López-Burgos et al. (2013) to create "cloud-free" SCA maps. Deng et al. (2015) combined MOD, MYD and
SNOWL SCE and AMSR2 SWE data and temporal filtering to create a daily "cloud-free" snow cover maps of
China. Combining different methods sequentially to remove clouds is also a way to create CGF products (Dariane
et al., 2017). Crowdsourcing by cross-country skiers combined with MODIS snow-cover products has also been
used to create daily CGF products (Kadlec and Ames, 2017).

A common method to reduce cloud cover on a daily snow map is to combine the daily Terra (MOD10A1) and Aqua
(MYD10A1) snow maps (see for example, Gao et al., 2010 & 2011;
Li et al., 2017; Paudel and Anderson, 2011; Thompson et al., 2015; Yu et al., 2016; Xu et al., 2017). Dong and
Menzel (2016) developed a multistep method including probability interpolation, to eliminate cloud cover using
combined Terra-Aqua MODIS snow-cover products. This takes advantage of the fact that the Terra and Aqua
satellite overpasses occur at different times of the day and, since clouds move, oftentimes more snow cover or non-
snow-covered land cover can be imaged and mapped using data from both satellites, as compared to using the Terra
or Aqua MODIS data alone. However this method of cloud clearing is of limited utility because changes in cloud
cover are typically small between Terra's 10:30 am local time equator crossing and Aqua's at 1:30 pm.

Percent reductions in cloud cover combining Terra and Aqua daily snow-cover data are highly variable and
dependent on many factors such as location, time of year, daily weather and cloud conditions, etc., and have been
reported to vary. A factor that impacts the quality of both the Aqua MODIS snow-cover and the cloud-cover
products, used to mask clouds, is that the critical 1.6 $\mu$m band used in both algorithms is non-functional on the Aqua
MODIS. As an example, for the western U.S. study area shown in Fig. 1, for 14 March 2012 and 19 March 2012,
using a snow-cover map that combined Terra and Aqua snow cover products, the MOD10 snow product showed
71.7 percent clouds while the combined Terra and Aqua products showed 67.0 percent for 14 March 2012; for
another date, 19 March 2012, MOD10 showed 71.8 percent clouds while the combined Terra/Aqua snow map
showed 68.4 percent. Combining the MOD and MYD snow maps definitely can reduce cloud cover but there are
issues with the Aqua snow maps (see below) and reliance on the continued availability of two nearly-identical
sensors is problematic.



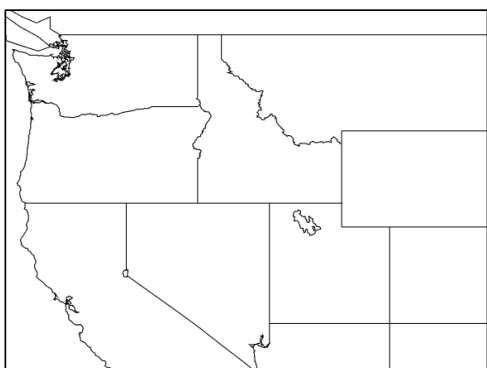


**Figure 1:** Study area covering all or parts of nine states in the western United States and part of southern Canada. The following
MODIS tiles were used to develop the composite: h08v04, h09v04, h10v04, h08v05, h09v05, h10v05.

Beginning in C6 the Quantitative Image Restoration (QIR) algorithm (Gladkova et al., 2012) has been used in the
Aqua MODIS snow algorithm to restore the lost data from the non-functional band 6 detectors so that the same
snow-cover mapping algorithm can be used in both Terra and Aqua. The cloud-mask algorithm for Terra uses
MODIS band 6 but the cloud-masking algorithm for the Aqua algorithm was adapted to use band 7 instead of band 6
for Collection 5 and earlier collections. This resulted in the Terra and Aqua algorithms providing different snow-
mapping results in many snow-covered areas due to the reduced accuracy of the Aqua algorithm. However, even in
C6 and C6.1 in which the QIR is employed, there are still more cloud/snow discrimination errors in the Aqua cloud-
mask algorithm as compared to the Terra algorithm. This results in more snow commission errors in MYD10L2
(Aqua) as compared to MOD10L2 (Terra). Because of the greater uncertainties inherent in snow mapping using
MYD10 algorithms for reasons mentioned above, and because any combined method using both Terra and Aqua
data is dependent on more than one sensor providing data, we do not recommend the MODIS Aqua SCE product to
be part of a planned MODIS-VIIRS ESDR for SCE. Additionally, since both the Terra and Aqua MODIS sensors
are well beyond their design lifetimes, it is not realistic to depend on both to provide data indefinitely into the future.
Fusion of ground based and satellite based snow observations is also an effective approach to "see beneath" clouds.
This method of cloud clearing is used by NOAA to develop the Interactive Multisensor Snow and Ice Mapping
System (IMS) SCE products (see Helfrich et al., 2007 and 2012).
Our objective is to generate the CGF snow maps daily in the normal operational processing stream of MODIS and
VIIRS snow products. The cloud-clearing method uses current day and/or recent previous day(s) of MODIS daily
snow-cover products to fill gaps created by cloud cover. If timeliness were not a constraint then interpolation of
snow cover over time, both on previous and future days, could be a part of a cloud-clearing algorithm, and would
increase the accuracy of the snow cover map on any given day.





**3 Methodology and Results**

The new standard CGF products, M*D10A1F and VNP10A1F, enable researchers to download and use cloud-free
MODIS and VIIRS daily snow maps along with quality-assurance (QA) data to assess uncertainties of the gap-
filling algorithm.  The daily MODIS Terra CGF SCE product is similar to MOD10A1 product but is cloud-free as
seen in Fig. 2.

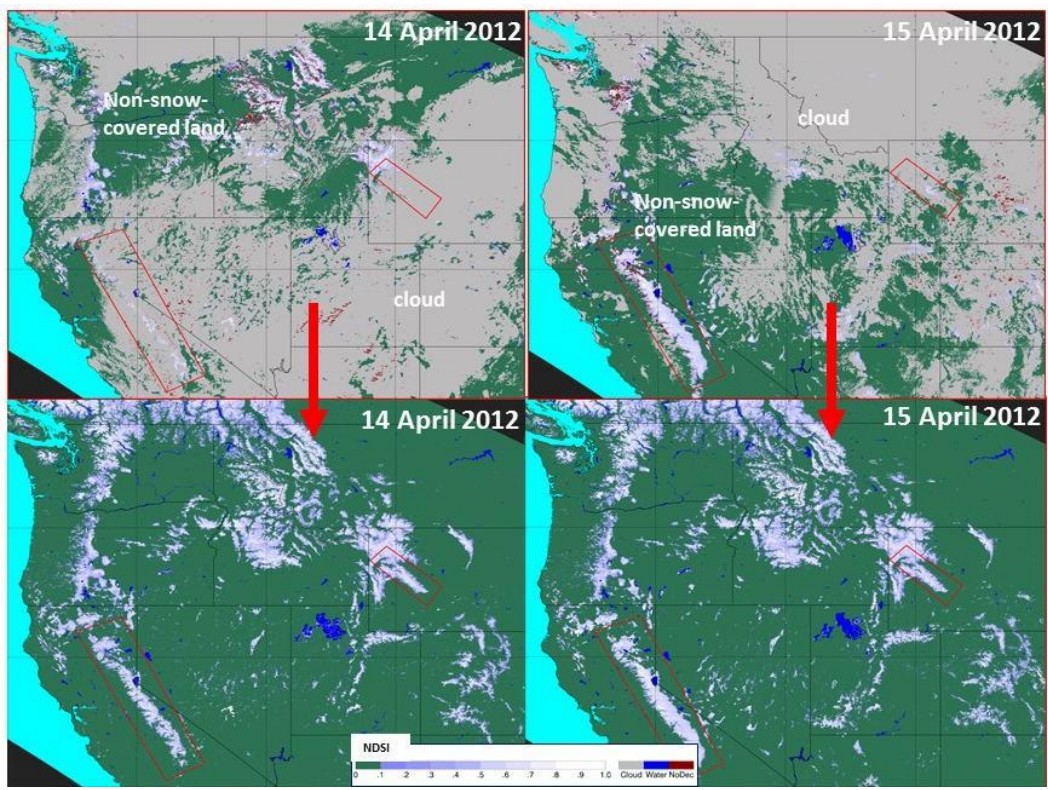


**Figure 2: Top Row -** Examples of the C6 MOD10A1 and the new Collection 6.1 MOD10A1F MODIS snow maps for a study
area in the western United States (see **Fig. 1**). **Top row**: MOD10A1 snow maps showing extensive cloud cover on 14 and 15
April 2012. **Bottom row**: MOD10A1F cloud-gap filled (CGF) maps corresponding to the MOD10A1 maps in the top row, also
for 14 and 15 April 2012.  Non-snow-covered land is green.  Regions of interest (ROI) containing the Sierra Nevada Mountains
in California and Nevada (109,575 km$^2$), and the Wind River Range in Wyoming (22,171 km$^2$), are outlined in red.

Though cloud-gap filling provides a cloud-free snow map every day, the accuracy of the snow observation depends
in part on the age of the observation, i.e., number of days since last cloud-free observation, thus information on
cloud persistence is included with each product.  The accuracy of the observation at the pixel level depends on the
snow-cover algorithm that includes cloud masking of the swath product, M*D10_L2, for MODIS and VNP10_L2



232 for VIIRS. The MODIS and VIIRS snow-cover swath products are gridded and mapped into the daily tiled products

233 which are input to M*D10A1F and VNP10A1F CGF algorithms.


235 For MODIS, inputs to the CGF algorithms are the current day M*D10A1 and the previous day M*D10A1F

236 products. The CGF daily snow map is created by replacing cloud observations in the current day M*D10A1 with

237 the most-recent previous cloud-free observation from the M*D10A1F (Hall et al., 2010; Riggs et al., 2018). The

238 algorithm tracks the number of days since the last cloud-free observation by incrementing the count of consecutive

239 days of cloud cover for a pixel. This is stored in the cloud-persistence count (CPC) data array. If the current day

240 observation is 'cloud' then the cloud count is one and is added to the CPC count from the previous day's

241 M*D10A1F and written to the current day's M*D10A1F algorithm. If the current day observation is 'not cloud,'

242 then the CPC is reset to zero in the current day's M*D10A1F CPC. If the CPC is 0, that means that the snow-cover

243 observation is from the current day. If the CPC for the current day is ≥1, that represents the count of days since the

244 last 'non-cloud' observation. On the day that the CGF mapping algorithm is initialized for a time series, for

245 example, 1 February 2012, the CGF snow-cover map is identical to the MODIS daily snow-cover map (M*D10A1)

246 and the cloud-persistence count (CPC) map will show zeros for non-cloud observations and ones for cloud

247 observations (Riggs et al., 2018). As the time series progresses a nearly-cloud-free snow map is produced on about

248 Day 5 in this example, on which the percent cloud cover is only 3.8 percent (Fig. 3), though it takes 24 days to

249 achieve a completely cloud-free map (not shown). The same method is used to develop the VNP10A1F CGF

250 products.

251

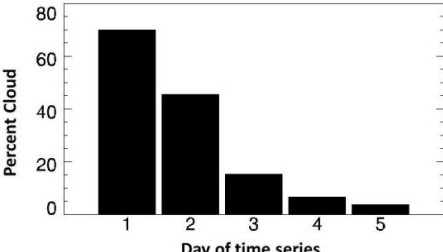

252

253 **Figure 3:** Percent cloud cover on a scene from the western United States (see location of the study area in Fig. 1). Note that the

254 percentage of cloud cover decreases dramatically in the first few days following the beginning of the CGF time series on 1

255 February 2012, herein denoted as Day 1. The percent cloud cover drops from about 75 percent on Day 1 to 3.8 percent on Day 5.

256

257 A CPC map is associated with each CGF snow map so that a user may determine the age of the snow observation of

258 each pixel. For each pixel, the uncertainty of the observation increases with time since the last clear view. To help a

259 user assess the accuracy of an observation, the count of consecutive days of cloud cover is incremented and stored as

260 QA in the CPC map that specifies how far back in time the observation was acquired. For example, for 19 March

261 2012, when CPC = 0, this means that the reported NDSI value for that pixel was acquired on 19 March 2012. When

262 CPC=1 this means that the reported NDSI pixel value is one day old, hence it was acquired on 18 March, and so on



(Fig. 4).  A user can decide how far back in time they would like to use an observation, and can easily develop a
unique CGF map, utilizing the CPC information that is most appropriate for their application.


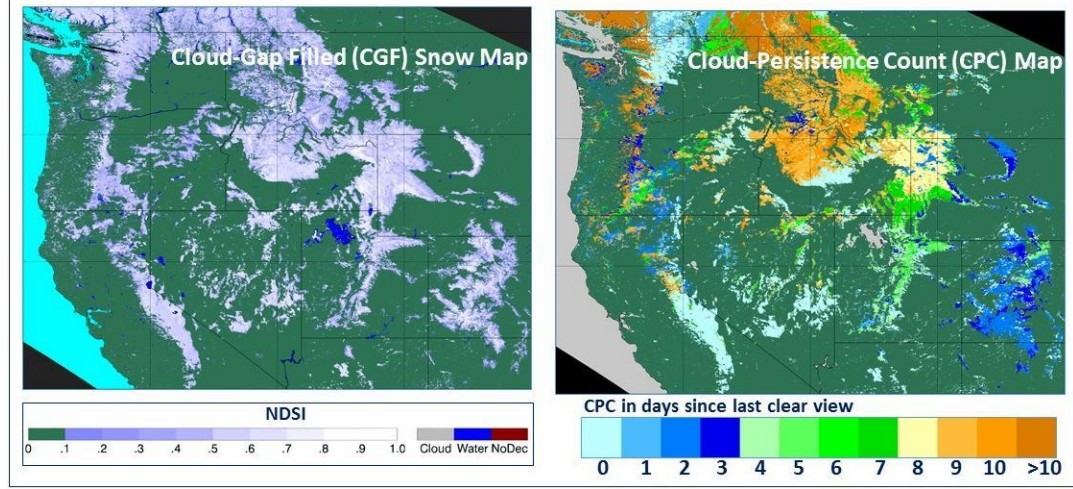



**Figure 4:**  Left - Cloud-gap filled (CGF) MOD10A1F snow map for 19 March 2012.  Right – Cloud-persistence count (CPC)
map from the quality assurance (QA) dataset for the 19 March CGF snow map seen at left.

The time series are started with the first day of acquisition for each mission, then reset when October 1$^{st}$ is reached.
The first days of the gap-filling time series for the Terra and Aqua MODIS CGF production are 24 February 2000
and 24 June 2002, respectively.  The first day of gap filling for the S-NPP VIIRS CGF production is the first day of
VIIRS data collection which is 21 November 2011.  With those exceptions, gap-filling sequences begin on the first
day of each water year, October 1$^{st}$.

The MODIS data-acquisition record is nearly continuous from the beginning of the missions however, there are brief
periods -- minutes to hours -- when either the Terra
[https://modaps.modaps.eosdis.nasa.gov/services/production/outages_terra.html] or Aqua
[https://modaps.modaps.eosdis.nasa.gov/services/production/outages_aqua.html] MODIS data were not acquired or
data were "lost."  In general, those outages have minimal effect on the snow-cover data record.  However there are
rare extended data outages of one to five days that have occurred, and may occur in the future.  The gap-filling
algorithms for both MODIS and VIIRS are designed to continue processing over daily or multi-day gaps in the data
record.  A missing day of MODIS or VIIRS NDSI snow-cover input is processed as if it were completely cloud
obscured so the previous day's CGF result is retained and the CPC is incremented by one.  Orbit gaps and missing
swath or scan line data within a tile are processed as a cloud observation with the previous good observation retained





and the CPC is incremented for the current day.  This provides a continuous data record for the CGF product. See
Riggs et al. (2018) for further details.
**3.1 Evaluation and Validation Analysis**
There are many ways to evaluate the uncertainties in the CGF snow-cover maps but only one way to validate the
maps.  The CGF maps can be compared with other daily snow-cover map products (e.g., NOAA IMS 4-km snow
maps Helfrich et al., 2007; 2012; Chen et al., 2012), with snow maps developed from higher-resolution maps such as
from Landsat and Sentinel and with reflectance images derived from satellite data.  This allows us to evaluate the
products but does not constitute validation.  The only way to validate the product is using NOAA snow depth data
https://gis.ncdc.noaa.gov/maps/ncei/summaries/daily as has been done for MOD10A1 (Collections 1 – 5) by many
authors (e.g., Brubaker et al., 2005; Chen et al., 2012).  However the density of meteorological stations is highly
variable.  Therefore the snow maps can only truly be validated where there is a dense network of meteorological
stations, though we can sometimes successfully interpolate between stations when stations are farther apart.
*Compare with NOAA snow depth data*.  Snow depths from NOAA snow depth data (e.g., see Fig. 5) can be overlain
on a MODIS CGF snow map as shown in Figs. 6 and 7.

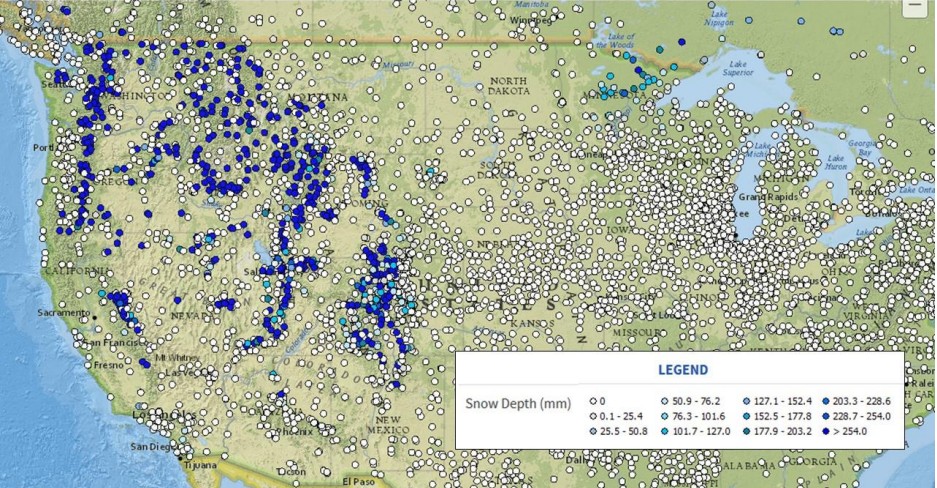

**Figure 5:** Snow depth (mm) from 16 April 2012 for part of the continental United States.  Source: NOAA National Climate Data
Center https://gis.ncdc.noaa.gov/maps/ncei/summaries/daily.
On 16 April 2012 the MODIS CGF map appears to map the location of snow cover very well in an ROI in Utah that
includes part of the Wasatch Range, based on NOAA snow-depth data indicating the presence of snow cover.  A





NASA WorldView true-color (corrected reflectance) MODIS Terra image is shown alongside a MODIS Terra CGF
snow map with NOAA snow depths superimposed on an ROI in south-central Utah (Fig. 6a, b & c).  There are no
other NOAA stations that report snow cover except the ones shown in Fig. 6b.  The dark blue and light blue circles
indicate snow depths of up to or >254.0 mm, and the white circle indicates a snow depth of 0.1 – 25.4 mm, revealing
that the MOD10A1F snow map accurately reflects the location of snow cover in this ROI.

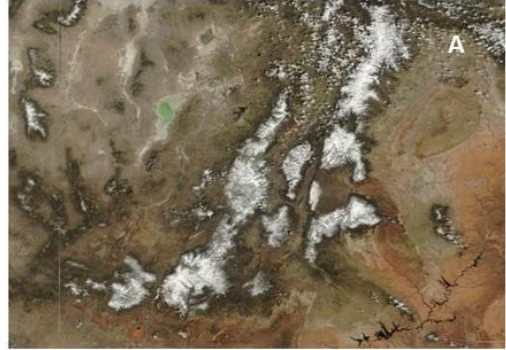

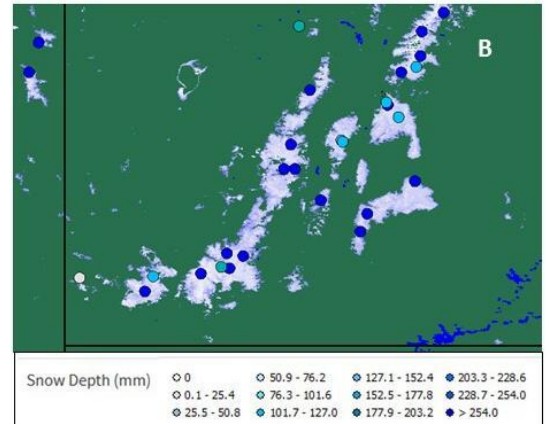

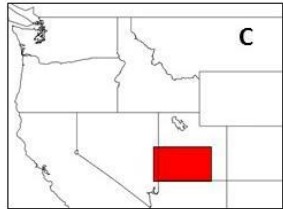


**Figure 6a:**  NASA WorldView true-color (corrected reflectance) MODIS Terra image of central Utah, including the southern
part of the Wasatch Range, acquired on 16 April 2012.  Fig. 6b. Snow depths from NOAA are mapped onto the MODIS Terra
CGF map, MOD10A1F, for 16 April 2012 for the same area shown in Fig. 6a.  Open circles indicate stations that report snow
depth, though none is visible in this snow map.  Fig. 6c.  Location map.
*3.11 Compare with higher-resolution images and derived snow maps*.  A good way to evaluate the accuracy of the
CGF SCE maps is to compare them with snow maps derived from higher-resolution sensors.  As an example, we
compare snow cover mapped in MODIS CGF snow-cover products with snow cover derived from Sentinel-2A
Multispectral Instrument (MSI) 30-m resolution images from the Harmonized Landsat Sentinel-2 (HLS) dataset
[https://hls.gsfc.nasa.gov/] (Claverie et al., 2018) as seen in for an ROI in Montana, in Fig. 7a, b & c.







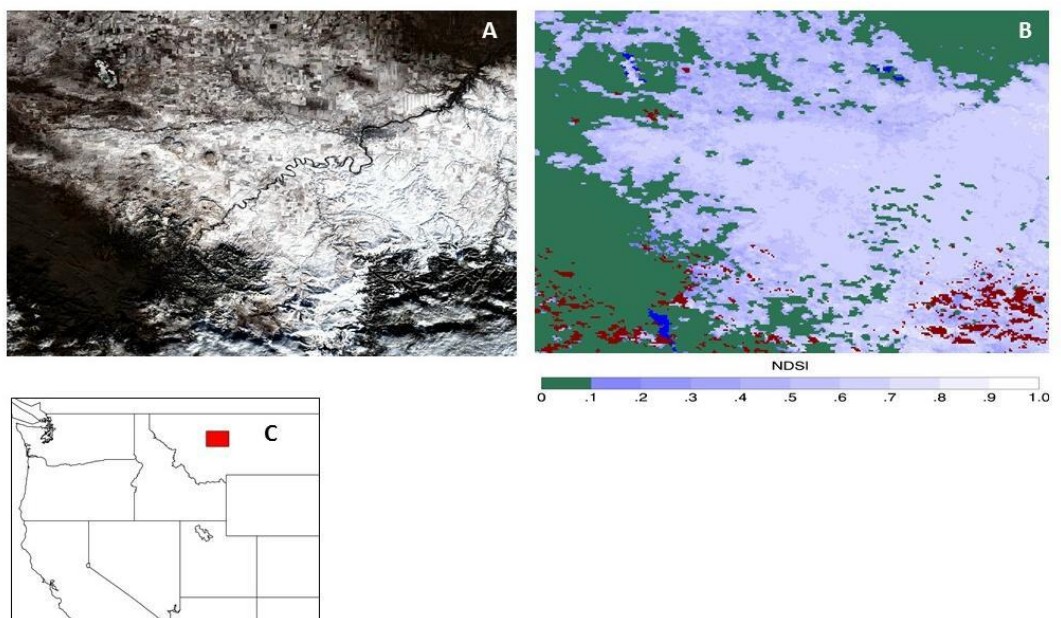

**Figure 7a:** Sentinel-2A 'true-color' image showing snow cover in shades of white and grey, acquired on 2 December 2016 for an ROI in the state of Montana. Black indicates non-snow-covered ground. Fig. 7b. The MOD10A1F cloud-gap-filled (CGF) snow map of the same area and on the same date as is shown in Fig. 7a. In the CGF snow map in Fig. 7b, snow is depicted in various shades of white and purple, corresponding to Normalized Difference Snow Index (NDSI) values. Pixels shown in red represent 'no decision' by the NDSI algorithm. Fig. 7c. The red box corresponds to the location of the images in Montana, shown in Fig. 7a and Fig. 7b.

Snow cover on 2 December 2016 may be seen on the Sentinel-2A image in shades of white and grey from this RGB composite image (bands 4, 3 and 2 (red (664.6 nm), green (559.8 nm) and blue (492.4 nm), respectively)) in Fig. 7a. Though the location of snow cover in the S2 image is visually very close to the snow cover depicted in shades of purple to white in the CGF snow map of Fig. 7b, there is not perfect correspondence. The point is to demonstrate the utility of high-resolution imagery to evaluate the CGF maps, not to perform a detailed and quantitative comparison.

*3.12 Effect of cloud cover on the accuracy of the CGF snow-cover maps*. The accuracy of the CGF snow decision in each pixel is influenced by cloud persistence, or the number of days of continuous cloud cover. This is because the algorithm updates the snow map under clear-sky conditions, or when there are breaks in cloud cover, according to the cloud mask. To demonstrate differences in cloud cover and thus to illustrate differences in CGF uncertainty, between two areas in the United States, we show the mean number of days of continuous cloud cover for a study area in the western U.S./northern Mexico and in the northeastern U.S./southeastern Canada for the month of February 2012 (Fig. 8a, b & c). Greater accuracy in snow-cover decisions for the CGF snow-cover product is





possible when there are more views of the surface as in the western U.S. (that includes the Sierra Nevada Mountains
ROI discussed earlier) vs. in part of the northeastern U.S. (Fig. 8a).  For example, for February 2012 the mean
number of days of continuous cloud cover on a per-grid cell basis in the northeastern U.S./southeastern Canada
(2.67 days) is greater than in the western U.S./northern Mexico (0.49 days) as seen in Fig. 8b.  Figs. 8a and 8b
demonstrate graphically that there were more views of the surface in the western study area as compared to the
eastern study area for the month of February 2012.  Thus the expectation is, that the accuracy of the CGF snow maps
at this time of year is higher in western U.S. study areas as compared to cloudier northeastern U.S. study areas.

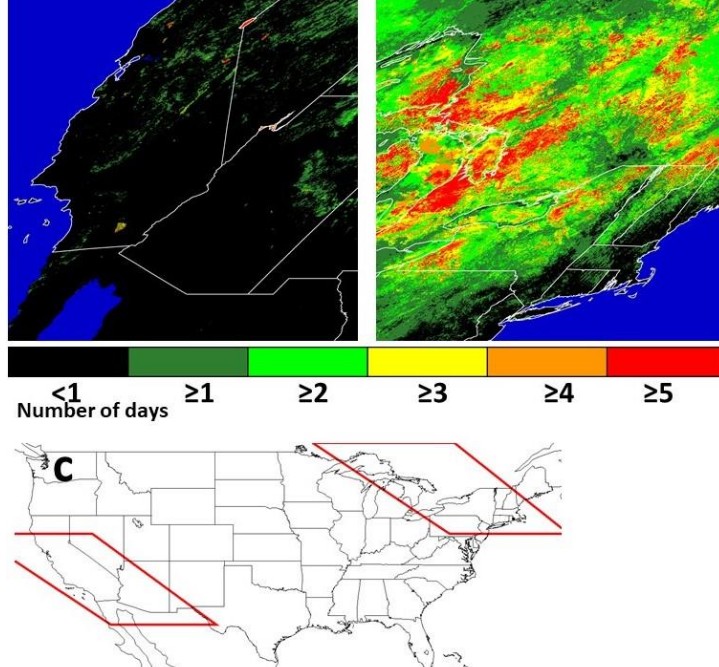

**Figure 8a and 8b:**  Maps showing the mean number of days of continuous cloud cover (a measure of cloud persistence) for
February 2012 derived from the MOD35 cloud mask used in the MOD10A1F snow-cover products.  Fig. 8a.  A study area in the
western U.S., extending into northern Mexico.  Fig. 8b.  A study area in the Northeastern U.S./southeastern Canada.  Fig. 8c.
Map showing the locations of the study areas shown in Figs. 8a and 8b.

Development of Environmental Science Data Records using Cloud-Gap Filled Snow Maps

*3.13 Comparison of Terra and Aqua MODIS snow maps for inclusion in an Earth Science Data Record (ESDR).*
We analyzed Terra and Aqua CGF snow maps and time-series plots to determine which maps are better suited to
being part of the SCE ESDR.  First we compared snow-map data from both Terra and Aqua from 1 February
through 30 April 2012 for ROIs including the Wind River Range, Wyoming, and the Sierra Nevada Mountains in
California and Nevada (see red rectangles in Fig. 2 for locations).  In the first few days of each time series, the CGF




algorithm is actively removing clouds from the daily maps, until both the Terra and Aqua daily maps are completely
cloud-free by approximately DOY 20 of the Wind River Range ROI time series and Day 10 of the Sierra Nevada
ROI time series as seen in Fig. 9. Pixels for which the algorithm provided "no decision" were excluded from the
analysis. The plots on the top row in Fig. 9 show the MODIS Terra and Aqua agreement of percent snow cover as
R=1.0, and Mean Bias=1.69 for the Wind River Range ROI time series and R=0.96 and Mean Bias=1.13 for the
Sierra Nevada ROI time series. Difference in percent clouds in each ROI (in which the difference = Terra minus
Aqua) reveals that the Aqua snow maps generally have more clouds than do the Terra snow maps.

Even when the "no-decision" pixels are excluded, there are still differences in Terra and Aqua cloud masking that
preclude the Terra and Aqua time series from being identical. This is especially notable from ~DOY 35 – 70 of the
Wind River Range time series (see top left graph in Fig. 9). This corresponds to a period with significant cloud
cover that is being mapped differently by the Terra and Aqua cloud masks (see bottom row in Fig. 9). Difference in
percent cloud cover by day for MODIS Terra minus Aqua for the ROI including the Wind River Range and the ROI
including the Sierra Nevada Mountains are shown in the bottom row of Fig. 9. The Aqua MODIS tends to have
more cloud cover during the study period than does the Terra MODIS.

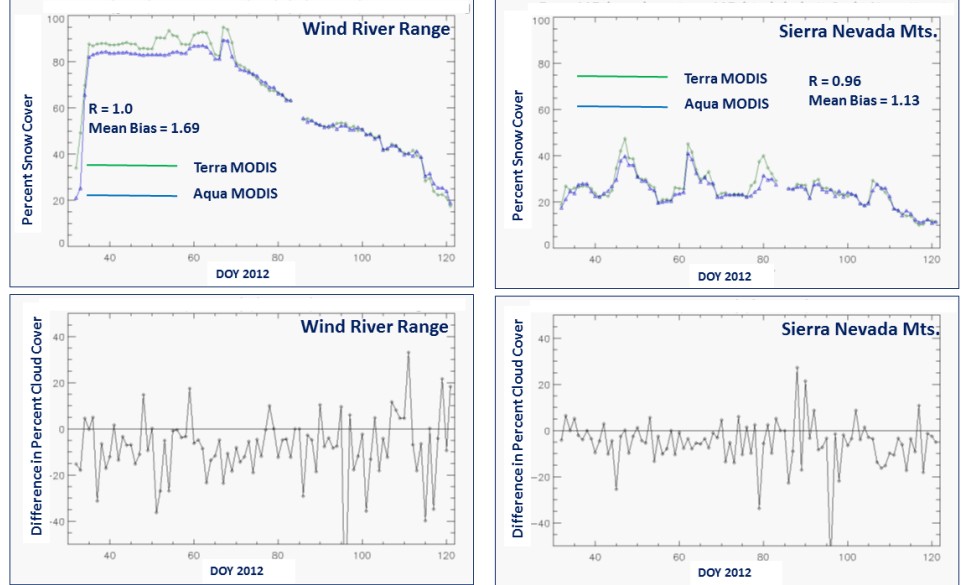


**Figure 9**: Top Row. Time-series plots of percent snow cover in a 22,171 km² scene (see location of the ROI that includes the
Wind River Range, Wyoming, in Fig. 2) and in a 109,575 km² scene (see ROI that includes the Sierra Nevada Mts., in Fig. 2)
using M*D10A1F snow-cover maps for a time series extending from 1 February through 30 April (DOY 32 – 121) 2012.
Bottom Row. Difference in percent cloud cover by day for MODIS Terra minus Aqua for the ROI including the Wind River
Range and the ROI including the Sierra Nevada Mountains, corresponding to the top panels, showing that the Aqua MODIS has
more cloud cover during the study period than does the Terra MODIS.






Though the percent snow cover on the Terra and Aqua snow maps is highly correlated in the example time series
shown in Fig. 9, there is also quite a bit of disagreement for example from about DOY 35 – 70 for the Wind River
Range.  Our analysis of both CGF snow maps indicates that the Terra MODIS snow map is superior for reasons that
are discussed below.

The primary reason for disagreement between the MODIS Terra and Aqua snow maps in C5 and earlier collections
is that the 1.6 µm channel (Band 6) on the Aqua MODIS sensor has some non-functioning detectors (MCST, 2014).
Other reasons include low illumination and terrain shadowing.  The reader is referred to the MODIS C5 User Guide
(Riggs et al., 2016) for more details concerning the effect of the non-functioning detectors on the Aqua snow-cover
maps in data collections prior to C6.

For C6, the MYD10A1 algorithm uses the Quantitative Image Restoration (QIR) of Gladcova et al. (2012) to correct
the band 6 radiances for the non-functioning detectors, and thereby to enable use of the same algorithm as is used for
the Terra MODIS.  Differences in cloud cover, and in cloud masking account for differences in snow-mapping
results between the C6 Terra and Aqua MODIS snow maps shown in Fig. 9.  The lower panels in Fig. 9 illustrate
differences in the cloud masking for Terra and Aqua for the 1 February – 30 April 2012 time series.

A specific example illustrating this can be seen on 26 April 2012 which was a day that had a large amount of clouds
in our study area of the western United States (Fig. 10).  The patterns of cloud cover in the false-color imagery (not
shown) of both MODIS Terra and Aqua show that the clouds are in the same shape of many of the 'no-decision'
regions on the Aqua CGF snow map.  The clouds are probably very cold (possibly with ice) on top of lower-level
clouds. The Aqua cloud mask fails to flag most of those clouds as 'certain cloud,' so they are processed as 'clear' in
the MYD10A1 snow algorithm, and 'no decision' is the result.  This is an outcome of the fact that band 6 is not used
in the Aqua cloud masking algorithm because of the non-functioning detectors.  Even though MYD10A1 uses the
QIR for C6, the C6 cloud masking algorithm does not.

This is a common problem with the C6 Aqua CGF snow maps, and the large number of 'no decision' pixels
resulting from the C6 cloud mask would affect the continuity of an ESDR.  For that reason, we have decided to use
the Terra CGF maps only, as part of the ESDR.





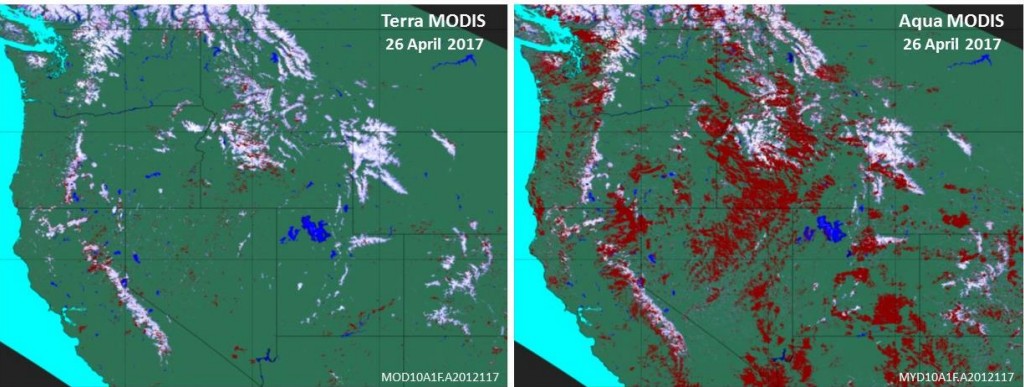


**Figure 10:** Terra MODIS (left) and Aqua MODIS (right) cloud-gap-filled (CGF) snow-cover maps from 26 April 2012. Note
that there are red pixels on both snow maps indicating 'no decision' by the algorithm, however there are many more red pixels on
the Aqua snow map, due largely to the inability of the Aqua MODIS cloud mask to identify large areas of cloud cover as 'certain
cloud.'

**4 Discussion and Conclusion**

Meltwater from mountain snowpacks provides hydropower and water resources to drought-prone areas such as the
western United States. Accurate snow measurement is needed as input to hydrological models that predict the
quantity and timing of snowmelt during spring runoff. SCE can be input to models to estimate snow-water
equivalent (SWE) which is the quantity of most interest to hydrologists and water management agencies.
Increasingly-accurate predictions save money because reservoir management improves as measurement accuracy of
SWE increases.

In this paper, we describe some of the applications and uncertainties of the C6.1 MODIS cloud-gap filled (CGF)
daily snow-cover map, M*D10A1F and the C2 VIIRS CGF snow-cover map, VNP10A1F. The objective of this
work is to produce a daily, cloud-free snow-cover product along with appropriate QA information that can be used
as the basis for an Earth Science Data Record (ESDR) of snow cover at moderate spatial resolution. Cloud-gap
filled snow-cover products from MODIS and VIIRS have all of the uncertainties of the original products, that
contain clouds, as well as additional uncertainties that are related to the age of the snow measurement. When using
the MODIS and VIIRS CGF products, a user can specify how far back in time they want to look, using the Cloud-
Persistence Count (CPC) which tells the age of the snow measurement in each pixel, and is available as part of the
product QA metadata for both the MODIS and VIIRS CGF snow-cover products. Uncertainty relating to cloud-gap
filling is greater in areas with frequent and persistent cloud cover during the snow season such as the northeastern
U.S. or WRR vs the Sierra Nevada Mountains.

It is difficult to validate the MODIS and VIIRS CGF (and other) snow maps. Absolute validation can be
accomplished using NOAA daily snow depth station data when available. We can also evaluate the product



accuracy by comparing the CGF with MODIS surface reflectance maps, higher-resolution maps such as derived
from Landsat and Sentinel and other snow maps.

Comparisons of Terra and Aqua CGF snow maps in C6 reveal many more "no-decision" pixels in the Aqua snow
maps, due to cloud masking, low illumination and terrain shadowing.  Because of non-functioning detectors in band
6, the Aqua cloud mask is less accurate than the Terra cloud mask.  The Terra and Aqua snow algorithms are the
same in C6 due to use of the Quantitative Image Restoration technique for Aqua, but the accuracy of the Terra
product is higher, and therefore the Terra MODIS CGF snow-cover maps of C6.1 are useful for development of an
ESDR and ultimately a CDR (combined with S-NPP VIIRS and other JPSS VIIRS-derived snow maps now and in
the future).

Time series of both the Terra and Aqua daily CGF snow-cover maps show pixels classified as 'no decision,' but on
the Aqua CGF maps, there are many more 'no decision' pixels on the Aqua maps.  Because of this issue with the
Aqua MODIS cloud masking, as detailed above, we do not recommend using the C6 Aqua MODIS snow maps as
part of an ESDR at this time.  In the future, if the Terra and Aqua cloud mask algorithms become more similar in
future re-processing of the cloud mask, this recommendation will be reassessed.

Snow cover is one of the Global Climate Observing System (GCOS) essential climate variables.  The distribution,
extent and duration of snow, along with knowledge of snowmelt timing are critical for characterizing the Earth's
climate system and its changes.  To augment the 53-year NOAA/Rutgers CDR of snow cover at 25-km resolution
which is valuable for climate and other studies, the MODIS/VIIRS moderate-resolution ESDR will be available at
500-m resolution and as such is useful for local and regional studies of snow cover and water resources, as well as
climate studies as the length of the record increases.

**Acknowledgements**

We would like to acknowledge support from NASA's Terrestrial Hydrology (grant # 80NSSC18K1674) and Earth
Observing Systems programs (grant # NNG17HP01C).  The *Sentinel*-2A satellite is operated by the European Space
Agency (ESA); a collaborative effort between ESA and the USGS provides a data portal for Sentinel-2A data
products.

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
