# Peer review of "Evaluation of MODIS and VIIRS Cloud-Gap Filled Snow Cover Products for production of an Earth Science Data Record 3"

_Hydrology and Earth System Sciences, 2019_

## Referee Comment (RC1) · Anonymous Referee #1 · 9 May 2019

General comments

The objective of this study is to describe algorithm for removal clouds in MODIS (Terra and Aqua) and VIIRS snow cover products. The aim is to develop/describe a method, which will generate cloud-free images of MODIS and VIIRS in operational practice.

The topic of the study is for sure within the scope of the journal. The manuscript reads well, however, significant novel scientific contribution is not clearly formulated nor demonstrated. It is not clear in which respect is described cloud filling method new, compared to existing approach. The idea of providing VIIRS snow cover product as alternative to MODIS is interesting, however the manuscript in its current form provide only very limited quantitative evaluation of the new features of presented method, about accuracy and comparison between the products (MODIS, VIIRS) and differences in

the efficiency of gap-filling method for these two (MODIS and VIIRS) products. This is the main limitation of the (current form of) study and this is the main reasons for recommending a substantial revision of the manuscript.

Specific comments

1) Abstract: l.27-28: " work is ongoing...". It will be interesting to see the results of such evaluation in the study. Otherwise the sentence is not very informative for the readers.

2) The study proposes VIIRS products to be an alternative to MODIS, but the cloud removal is presented only for MODIS. A thorough comparison with VIIRS will be interesting to see.

3) The novelty of the objectives is not clearly formulated. The wording as "to describe...", "to discuss..." does not clearly indicate quantitative contribution of the study.

4) Section 2.4: this section is interesting and shows different methods used for cloud removal in the past. However some organised synthesis of the approaches will be useful here. E.g. stratification of approaches according to different assumptions (e.g. temporal, spatial filters, snow line, multi-sensor combination, etc.).

5) Why only temporal filter is considered for gap-filling method? During snowmelt, snow-line approach or some kind of spatial filter can be more efficient.

6) The results show only few examples which does not allow to see clearly if the results are robust and general. More thorough analysis (longer time periods, seasonal evaluation, larger/different regions) will allow to draw much more robust findings.

---

## Referee Comment (RC2) · Anonymous Referee #2 · 25 May 2019

The manuscript provides an overview of MODIS snow products development, including the newest snow gap-filled (CGF) maps, efforts related to their validation and clouds removal, describes the methodology of daily snow-gap-filled maps preparation, presents several examples from the western and eastern USA and concludes that the Aqua MODIS snow gap-filled maps should not be used as the basis of Environmental Science Data Record (ESDR). I recommend major revision and have the following comments:

1. Objectives, novelty -Manuscript authors are among the leading scientists in MODIS snow products development, evaluation and utilization. However, the objectives of this manuscript are not clear to me. The manuscript feels more like a collection of ideas and examples than a systematic exploration/summary of the MODIS CGF snow cover

products advantages and uncertainties (manuscript title). At the end of the introduction it is mentioned "...In this paper, we describe the MODIS Terra and AQUA CGF algorithm, data products and uncertainties...." and "we also discuss the development of a moderate-resolution ESDR of SCE and using MODIS and VIIRS standard snow-cover maps". The latter is in my opinion not true, because the ESDR is just mentioned in the manuscript (its development is not discussed) and the same holds for VIIRS (it is not discussed, just mentioned that it exists and should extend the data through the 2030s). The CGF algorithm was already described in Hall et al. (2010). While the manuscript is informative, better indication of its novelty would be helpful.

2. Study area and period - I would welcome an explanation of study area and period selection. Why were particular study area and regions of interest selected? Fig. 1 presents western United states and part of southern Canada as the study area, but in the abstract and elsewhere, the northeastern United States are mentioned as well. Do the selected areas allow evaluation of uncertainties related to some issues mentioned in the text, e.g. rapid snow disappearance during the snowmelt or melting of the newly fallen snow during the cloudy periods? Figs 2,4,5,6 present data from spring 2012 (why namely that year?) while Fig. 7 shows the comparison of MODIS and Sentinel images for December 2016 (because the Sentinel data did not yet exist in 2012?).

3. Methodology and results - Because the objectives are not stated clearly, the methodology is in my opinion confusing as well. Lines 235-247 describe how are daily CGF maps created (methodology), but then the result on how quickly was a nearly cloud-free map obtained (in other areas and on other dates it can probably be achieved later or even earlier, a more systematic exploration would be interesting) is given. This result is followed by continuation of methodology (uncertainty based on the cloud persistence count and how the CPC is recorded). I recommend division of Methodology and Results. I agree that validation of the satellite data is only possible by comparison with measurements. The manuscript presents validation against the NOAA snow depth data provided by the dense network of meteorological stations. Such networks

are not available in other countries. Can we trust that the CGF maps are valid also in those parts of the world where the network density does not allow detailed validation? I would welcome a comment on this. Is it possible to conclude that Aqua snow maps tend to have have more clouds than the Terra snow maps globally? Can it not be only the case in some areas while in others it would be vice versa?

4. Discussion and conclusions - I appreciate the note that CGF snow products have all uncertainties of the original products as well as additional uncertainties that are related to the age of the snow measurement (l. 445-447).

Minor comment - text in lines 52-56 could be omitted.

---

## Author Comment (AC1) · 30 Jul 2019

**Response to Referee #1's comments**

**Authors responses are shown in bold font.**

**Authors:  We would like to thank both referees for their helpful and constructive comments and thorough reviews.  We have improved the flow of the paper and added more information about the VIIRS snow maps and quantitative comparisons between the new MODIS and VIIRS cloud-gap filled snow maps.  We have emphasized the novelty of the work as requested by both referees.  We have also re-stated our objectives as requested by both referees, and changed the title to better reflect the content of the paper.**

**Referee #1**

Referee:  General comments
The objective of this work is to describe algorithm for removal clouds in MODIS (Terra and Aqua) and VIIRS snow cover products. The aim is to develop/describe a method, which will generate cloud-free images of MODIS and VIIRS in operational practice.

**Authors:  We now have improved our statement of objectives.  We do not develop a method in this paper because the method has already been developed.  We now more clearly state our objective as follows:**
**"The objective of this paper is to introduce the new MODIS and VIIRS standard CGF daily SCE products and to provide preliminary evaluation of uncertainties in the gap-filling methodology so the products can be used as the basis for a moderate-resolution Earth Science Data Record (ESDR) of SCE."**

Referee: The topic of the study is for sure within the scope of the journal. The manuscript reads well, however, significant novel scientific contribution is not clearly formulated nor demonstrated. It is not clear in which respect is described cloud filling method new, compared to existing approach.

**Authors:  We now describe clearly the novelty of this work as compared to earlier work (e.g., Hall et al., 2010).  The Hall et al. (2010) paper described a gap-filling technique, but the basic technique outlined in that paper was not implemented until recently.  And the algorithms that have been implemented are much more complex than the one described in Hall et al. (2010).**

**We introduce the CGF products derived from the MODIS and S-NPP VIIRS.  These products have not been introduced previously in the peer-reviewed literature.  In the present paper, we describe new cloud-gap filled (CGF) MODIS and VIIRS algorithms and products (targeting hydrologists, climate scientists and modelers) that will become available to the scientific community in the fall of 2019.  These products include quality-assurance (QA) information that can be manipulated by a user to develop a unique product that can be tuned to a user's particular study area.  This new and unique capability has not**

been possible using earlier MODIS snow products.  Uncertainties of the products are addressed in terms of their use in the hydrological community.

Also new is the finding that the Terra MODIS CGF snow product is superior to the Aqua MODIS CGF snow product, and the reason for this is discussed in the revised paper. Following on from this, we make the case that an Earth Science Data Record (ESDR) can be developed to begin in 2000 using Terra MODIS data, and continue through the present and beyond using VIIRS and JPSS data.

Additionally, new comparisons between the Terra MODIS and S-NPP VIIRS CGF snow products have been added: the comparisons are performed for a time series consisting of three months in the winter/spring of 2012.  We have not previously published Terra MODIS vs. S-NPP CGF snow-map comparisons in the peer-reviewed literature. Furthermore, we are not aware of anyone else doing such comparisons.  A new Figure 9 has been added as shown below:

[Figure]

**Figure 9:**  Time series showing differences in snow-cover extent (SCE) derived from Terra MODIS and S-NPP VIIRS cloud-gap filled (CGF) snow maps for a nearly 3-month period extending from 4 February – 30 April 2012.  Though the time series began on 1 February, snow-cover extent from 1 – 3 February snow cover is not shown because, in this example, the gap-filling algorithm was started on 1 February had not filled most of the gaps from clouds until 4 February.

**Figure 2 has been revised to demonstrate differences/similarities between the MODIS and VIIRS snow-cover products.  See below.**

[Figure]

**Figure 2:** Examples of MODIS and VIIRS standard and cloud-gap filled (CGF) snow maps on 14 April 2012 for a study area in the western United States/southwestern Canada (see **Fig. 1**). **Top left**: MODIS MOD10A1 C6.1 snow map showing extensive cloud cover on 14 April 2012. **Top right**: VIIRS VNP10A1 C1 snow map also showing extensive cloud cover on 14 April 2012. **Bottom left**: MOD10A1F C6.1 CGF map corresponding to the MOD10A1 snow map in the top row, also for 14 April 2012. **Bottom right**: VNP10A1F CGF map corresponding to the VNP10A1 snow map in the top row, also for 14 April 2012. In all of the snow maps, non-snow-covered land is green. Regions of interest containing the Sierra Nevada Mountains in California and Nevada (109,575 km$^2$), and the Wind River Range in Wyoming (22,171 km$^2$), are outlined in red on the MODIS snow maps. The following MODIS tiles were used to develop the MODIS composites: h08v04, h09v04, h10v04, h08v05, h09v05, h10v05. Each VIIRS swath that included coverage of this study area was composited to create a daily map, then the daily maps were used to create the VNP10A1F snow map for 14 April 2012.

**The following findings are new:**

- **Differences in cloud masking between MODIS and VIIRS affect the snow map results of both the standard product and the new CGF snow maps;**
- **Differences in cloud masking between the Terra MODIS and the Aqua MODIS affect the snow map results of both the standard product and the new CGF snow maps;**
- **The Terra MODIS snow maps are superior to the Aqua MODIS snow maps in C6 and C6.1;**
- **Development of an Earth Science Data Record that uses Terra MODIS and S-NPP VIIRS CGF snow maps is introduced.**

**In earlier work (e.g., Hall et al., 2010), we described a methodology that would provide daily, cloud-free snow maps globally using a gap-filling method with the daily climate modeling grid (CMG) composited products at 5-km resolution. Subsequent work (e.g., Hall et al. 2015) demonstrated the utility of this method applied to the daily tiled products for a study area in the Wind River Range of Wyoming. This gap-filling method, described in Hall et al. (2010), has been adopted as the basic algorithm of the standard product, M\*D10A1F, using the daily gridded products of individual observations at 500 m resolution and including *many* enhancements to the basic algorithm that was described in Hall et al. (2010).**

**In short, this paper is very timely because MODIS C6.1 and VIIRS C2 products are due to become available during the fall of 2019 and there is a great deal of information in this paper, especially in its revised form, that will be useful to users of the products.**

Referee: The idea of providing VIIRS snow cover product as alternative to MODIS is interesting, however the manuscript in its current form provide only very limited quantitative evaluation of the new features of presented method, about accuracy and comparison between the products (MODIS, VIIRS) and differences in the efficiency of gap-filling method for these two (MODIS and VIIRS) products. This is the main limitation of the (current form of) study and this is the main reasons for recommending a substantial revision of the manuscript.

**Authors: We appreciate this salient comment and have added quantitative evaluation of the Terra MODIS vs. S-NPP VIIRS CGF products. We developed a time series for a three-month period in the winter/spring of 2012. Please see revised Figure 2 and new Figure 9. Results show that the products agree well, though the VIIRS CGF maps show slightly more snow cover than do the Terra MODIS CGF maps. This is probably due to the fact that the VIIRS maps show fewer clouds than the Terra MODIS CGF maps show.**

Referee's specific comments:
Referee: 1) Abstract: l.27-28: " work is ongoing: : :". It will be interesting to see the results of such evaluation in the study. Otherwise the sentence is not very informative for the readers.

**Authors: We agree with this comment and have improved the wording in the abstract of the revised paper. We have now done additional work to quantify some of the uncertainties in the products, including some potential uncertainties in an ESDR that will use both the Terra MODIS and the S-NPP VIIRS.**

Referee: 2) The study proposes VIIRS products to be an alternative to MODIS, but the cloud removal is presented only for MODIS. A thorough comparison with VIIRS will be interesting to see.

**Authors: We cannot provide a thorough comparison of MODIS CGF and VIIRS CGF because the products are not yet available to download from NSIDC. Using swath data, however, we created a time series of MODIS CGF products for 2012. And, to address this**

comment, we have recently created a 3-month time series of VIIRS cloud-gap filled snow-cover extent (SCE) products and have plotted them along with the daily values (in km$^2$) with the Terra MODIS CGF.  This is shown in the new Figure 9, above.  Cloud cover on a single MODIS and VIIRS scene comparison is also shown in the revised Figure 2.

Furthermore, the revised Figure 3, shown below, demonstrates that the VIIRS product starts out with fewer clouds than does the Terra MODIS product, and clears the clouds faster according to the CGF algorithm.

[Figure]

**Figure 3:**  Percent cloud cover in a Terra MODIS (MOD10A1F) and an S-NPP VIIRS (VNP10A1F) time series of snow-cover maps for the western United States study area (see location in Fig. 1). Note that the percentage of cloud cover decreases dramatically in the first few days following the 4 February 2012 initiation of the CGF time series, denoted here as Day 1.

Creating time series before availability of the products through NSIDC (to occur in the fall of 2019) is computationally intensive.  This is the reason that we have not created multiple long time series.

The bottom line is that it is not possible for us to perform a thorough comparison, but we now include a graph showing a comparison of three months of both MODIS and VIIRS CGF daily snow maps.  The comparisons also demonstrate a great amount of agreement in SCE between the MODIS and VIIRS CGFs which is desirable for development of an ESDR.  This is quantified for the time period shown.

Referee: 3) The novelty of the objectives is not clearly formulated. The wording as "to describe: : :", "to discuss: : :" does not clearly indicate quantitative contribution of the study.

Authors: We have improved the description of the clarity of the objectives.  As also provided above, the following revised statement of our objectives appears in the Abstract and in the Introduction:
"The objective of this paper is to introduce the new MODIS and VIIRS standard CGF daily SCE products and to provide preliminary evaluation of uncertainties in the gap-filling methodology so the products can be used as the basis for a moderate-resolution Earth Science Data Record (ESDR) of SCE."

Referee: 4) Section 2.4: this section is interesting and shows different methods used for cloud removal in the past. However some organised synthesis of the approaches will be useful here. E.g. stratification of approaches according to different assumptions (e.g. temporal, spatial filters, snow line, multi-sensor combination, etc.).

**Authors: We have greatly shorted this section based on this comment. Since the CGF method to provide daily, cloud-free SCE maps for MODIS and VIIRS has already been determined and the products are starting to be produced (final production and release will begin in the fall of 2019), other cloud-clearing methods are somewhat irrelevant to the present work. However, we don't want to ignore other work that has been done to create daily cloud-reduced or cloud-free snow maps because there are many effective and novel cloud-clearing methods and much work has been done. Other works detailing methods that are also very useful for cloud clearing are important and are still cited though the entire section has been shortened. The organization of this section has been vastly improved.**

Referee: 5) Why only temporal filter is considered for gap-filling method? During snowmelt, snow-line approach or some kind of spatial filter can be more efficient.

**Authors: There are many other useful methods of gap filling, but the method described in our paper is the method that is used to develop the new product that will be available starting this summer or fall. We are beyond the point where different methods can be considered since the new algorithm uses the CGF method, all of the programming has been completed by the MODIS Project and the products will be available soon. It is too late to change the algorithm for Collection 6.1.**

Referee: 6) The results show only few examples which does not allow to see clearly if the results are robust and general. More thorough analysis (longer time periods, seasonal evaluation, larger/different regions) will allow to draw much more robust findings.

**Authors: We agree with this comment, but we are unable to do a thorough and global analysis because the product is not yet being produced by the MODIS and VIIRS Projects. When processing starts, the product will be downloadable through the National Snow and Ice Data Center starting in the fall of 2019.**

**In order to develop a time series in this pre-production phase, we need to do a considerable amount of programming. We've done this by developing a Terra MODIS CGF SCE time series for the western U.S. data set for 2012. For this revised version of the paper, and in response to this and other comments, we ran a 3-month time series using VIIRS SCE maps (see Figure 9). Running a CGF time series is computationally burdensome, and therefore a comprehensive, global analysis cannot be accomplished until after the official MODIS processing begins. Even after production begins it will take many months until the complete MODIS and VIIRS time series (from 2000 to present for MODIS and from 2011 to present for VIIRS) can be processed. Complete processing is likely to occur sometime in the year 2020 for both the MODIS C6.1 and VIIRS C2 CGF SCE products.**

**In short, processing will not be complete in a time frame that is reasonable for providing the revisions to this paper. And it is important that this paper be published so that users of the new products will have the information contained in this paper when the products first become downloadable from NSIDC in the fall of 2019.**

---

## Author Comment (AC2) · 30 Jul 2019

**Response to Referee #2's comments**

**Authors responses are shown in bold font.**

**Authors:  We would like to thank both referees for their helpful and constructive comments and thorough reviews.  We have improved the flow of the paper and added more information about the VIIRS snow maps and quantitative comparisons between the new MODIS and VIIRS cloud-gap filled snow maps.  We have emphasized the novelty of the work as requested by both referees.  We have also re-stated our objectives as requested by both referees, and changed the title to better reflect the content of the paper.**

**Referee #2**

Referee:  The manuscript provides an overview of MODIS snow products development, including the newest snow gap-filled (CGF) maps, efforts related to their validation and clouds removal, describes the methodology of daily snow-gap-filled maps preparation, presents several examples from the western and eastern USA and concludes that the Aqua MODIS snow gap-filled maps should not be used as the basis of Environmental Science Data Record (ESDR).

Referee:  I recommend major revision and have the following comments:
1. Objectives, novelty -Manuscript authors are among the leading scientists in MODIS snow products development, evaluation and utilization. However, the objectives of this manuscript are not clear to me. The manuscript feels more like a collection of ideas and examples than a systematic exploration/summary of the MODIS CGF snow cover products advantages and uncertainties (manuscript title).

**Authors:  We agree and have restated the objectives and clarified the novelty of the work in the revised manuscript.  Below you will find the new statement of the objectives (that now appears in the Abstract and in the Introduction) and a discussion of novelty of the work.  Since Referee #1 made the same comment, we are using a response that is very similar to the one we used to respond to Referee #1 regarding these points.**

**Authors:  We have improved both our 1) statement of the objectives of the work, and 2) our identification of its scientific novelty as follows:**

    **1)  The following was added to the Abstract and to the Introduction:**
**"The objective of this paper is to introduce the new MODIS and VIIRS standard CGF daily SCE products and to provide preliminary evaluation of uncertainties in the gap-filling methodology so the products can be used as the basis for a moderate-resolution Earth Science Data Record (ESDR) of SCE."**

    **2)  We now describe clearly the novelty of this work as compared to earlier work (e.g., Hall et al., 2010):**

We introduce the CGF products derived from the MODIS and S-NPP VIIRS. These products have not been introduced previously in the peer-reviewed literature. In the present paper, we describe new cloud-gap filled (CGF) MODIS and VIIRS algorithms and products (targeting hydrologists, climate scientists and modelers) that will become available to the scientific community in the fall of 2019. These products include quality-assurance (QA) information that can be manipulated by a user to develop a unique product that can be tuned to a user's particular study area. This new and unique capability has not been possible using earlier MODIS snow products. Uncertainties of the products are addressed in terms of their use in the hydrological community.

In the present paper, we describe new cloud-gap filled (CGF) MODIS and VIIRS algorithms and products (targeting hydrologists, climate scientists and modelers) that will become available to the scientific community in the fall of 2019. These products include quality-assurance (QA) information that can be manipulated by a user to develop a unique product that can be tuned to a user's particular study area. This new and unique capability has not been possible using earlier MODIS snow products. Uncertainties of the products are addressed in terms of their use in the hydrological community.

Also new is the finding that the Terra MODIS CGF snow product is superior to the Aqua MODIS CGF snow product, and the reason for this is discussed. Following on from this, we make the case that an Earth Science Data Record (ESDR) can be developed to begin in 2000 using Terra MODIS data, and continue through the present and beyond using VIIRS and JPSS data. New comparisons between the Terra MODIS and S-NPP VIIRS CGF snow products have been added: the comparisons are performed for a time series consisting of three months in the winter/spring of 2012. We have not previously published Terra MODIS vs. S-NPP CGF snow-map comparisons in the peer-reviewed literature. Furthermore, we are not aware of anyone else doing such comparisons.

Additionally, new comparisons between the Terra MODIS and S-NPP VIIRS CGF snow products have been added: the comparisons are performed for a time series consisting of three months in the winter/spring of 2012. We have not previously published Terra MODIS vs. S-NPP CGF snow-map comparisons in the peer-reviewed literature. Furthermore, we are not aware of anyone else doing such comparisons. A new Figure 9 has been added as shown below:

[Figure]

**Figure 9:** Time series showing differences in snow-cover extent (SCE) derived from Terra MODIS and S-NPP VIIRS cloud-gap filled (CGF) snow maps for a nearly 3-month period extending from 4 February – 30 April 2012. Though the time series began on 1 February, snow-cover extent from 1 – 3 February snow cover is not shown because, in this example, the gap-filling algorithm was started on 1 February had not filled most of the gaps from clouds until 4 February.

**Figure 2, below, has been revised to demonstrate differences/similarities between the MODIS and VIIRS snow-cover products.**

[Figure]

**Figure 2:** Examples of MODIS and VIIRS standard and cloud-gap filled (CGF) snow maps on 14 April 2012 for a study area in the western United States/southwestern Canada (see **Fig. 1**). **Top left**: MODIS MOD10A1 C6.1 snow map showing extensive

cloud cover on 14 April 2012. **Top right**: VIIRS VNP10A1 C1 snow map also showing extensive cloud cover on 14 April 2012. **Bottom left**: MOD10A1F C6.1 CGF map corresponding to the MOD10A1 snow map in the top row, also for 14 April 2012. **Bottom right**: VNP10A1F CGF map corresponding to the VNP10A1 snow map in the top row, also for 14 April 2012. In all of the snow maps, non-snow-covered land is green. Regions of interest containing the Sierra Nevada Mountains in California and Nevada (109,575 km$^2$), and the Wind River Range in Wyoming (22,171 km$^2$), are outlined in red on the MODIS snow maps. The following MODIS tiles were used to develop the MODIS composites: h08v04, h09v04, h10v04, h08v05, h09v05, h10v05. Each VIIRS swath that included coverage of this study area was composited to create a daily map, then the daily maps were used to create the VNP10A1F snow map for 14 April 2012.

**The following findings are new:**

- **Differences in cloud masking between MODIS and VIIRS affect the snow map results of both the standard product and the new CGF snow maps;**
- **Differences in cloud masking between the Terra MODIS and the Aqua MODIS affect the snow map results of both the standard product and the new CGF snow maps;**
- **The Terra MODIS snow maps are superior to the Aqua MODIS snow maps in C6 and C6.1;**
- **Development of an Earth Science Data Record that uses Terra MODIS and S-NPP VIIRS CGF snow maps is introduced.**

**In earlier work (e.g., Hall et al., 2010), we described a methodology that would provide daily, cloud-free snow maps globally using a gap-filling method with the daily climate modeling grid (CMG) composited products at 5-km resolution. Subsequent work (e.g., Hall et al. 2015) demonstrated the utility of this method applied to the daily tiled products for a study area in the Wind River Range of Wyoming. This basic gap-filling method, described in Hall et al. (2010), has been adopted as the basic algorithm of the standard product, M\*D10A1F, using the daily gridded products of individual observations at 500 m resolution and including *many* enhancements to the basic algorithm that was described in Hall et al. (2010).**

**In short, this paper is very timely because MODIS C6.1 and VIIRS C2 products are due to become available during the fall of 2019 and there is a great deal of information in this paper, especially in its revised form, that will be useful to users of the products.**

Referee: At the end of the introduction it is mentioned "...In this paper, we describe the MODIS Terra and AQUA CGF algorithm, data products and uncertainties...." and "we also discuss the development of a moderate-resolution ESDR of SCE and using MODIS and VIIRS standard snow-cover maps". The latter is in my opinion not true, because the ESDR is just mentioned in the manuscript (its development is not discussed) and the same holds for VIIRS (it is not discussed, just mentioned that it exists and should extend the data through the 2030s). The CGF algorithm was already described in Hall et al. (2010). While the manuscript is informative, better indication of its novelty would be helpful.

**Authors: We agree that we should have explicitly stated the novelty of this new work and we have now done that. Also please see discussion of novelty above.**

**We also now have images, snow maps and a graph showing VIIRS data as discussed in response to Referee #1's comments (new or revised Figures 2, 3 & 9).**

**In regard to the ESDR, we show an example of the Terra MODIS – VIIRS data continuity in the revised manuscript which is an important step towards development of an ESDR. We agree that an ESDR is not developed and have changed wording to suggest that the CGF products could be used as the basis of an ESDR. In this work we discuss some of the issues relevant to development of an ESDR such as continuity between products from different sensors and discuss differences such as that the VIIRS maps show slightly more snow than do the Terra MODIS maps because there are generally fewer clouds mapped by the VIIRS cloud mask.**

Referee: 2. Study area and period - I would welcome an explanation of study area and period selection. Why were particular study area and regions of interest selected? Fig. 1 presents western United states and part of southern Canada as the study area, but in the abstract and elsewhere, the northeastern United States are mentioned as well.

**Authors: In the revised manuscript in Section 4, we have included the following: "To enable some early evaluation of the products we produced CGF Terra and Aqua MODIS time series of areas in the western U.S. and in the northeastern U.S. and southeastern Canada. Here we provide evaluation and some validation for study areas in the western U.S. and a study area in the northeastern U.S./southeastern Canada. We also look at regions of interest (ROI) within our primary western U.S./southwestern Canada study area shown in Fig. 1. We selected the year 2012 for the time series because both MODIS and VIIRS data were available in that year. Comprehensive global validation studies will not be possible to perform until the data sets are released through NSIDC and the entire MODIS and VIIRS records have been processed. This will take several months following initial release of the data; the full data records should be available in 2020."**

Referee: Do the selected areas allow evaluation of uncertainties related to some issues mentioned in the text, e.g. rapid snow disappearance during the snowmelt or melting of the newly fallen snow during the cloudy periods?

**Authors: This large area in the western U.S. was selected because we are familiar with the area. The time series includes rapid snow disappearance during snowmelt and melting of newly-fallen snow. However we did not focus on those issues during our analysis.**

Referee: Figs 2,4,5,6 present data from spring 2012 (why namely that year?) while Fig. 7 shows the comparison of MODIS and Sentinel images for December 2016 (because the Sentinel data did not yet exist in 2012?).

**Authors: There was nothing special about the year 2012, or even 2016. These figures are examples. A random selection of dates during the snow season was selected. Sentinel data is available starting from about July 2015, so a 2016 winter image was used.**

Referee: 3. Methodology and results - Because the objectives are not stated clearly, the methodology is in my opinion confusing as well. Lines 235-247 describe how are daily CGF maps created (methodology), but then the result on how quickly was a nearly cloudfree map obtained (in other areas and on other dates it can probably be achieved later or even earlier, a more systematic exploration would be interesting) is given.

**Authors: The objectives are now stated clearly. Please see author comments above. We've included in this revision, a new graph (revised Fig. 3) that compares the MODIS and VIIRS CGF snow maps in terms of how long each time series takes to remove clouds. The new Figure 3 is shown below.**

[Figure]

**Figure 3:** Percent cloud cover in a Terra MODIS (MOD10A1F) and an S-NPP VIIRS (VNP10A1F) time series of snow-cover maps for the western United States study area (see location in Fig. 1). Note that the percentage of cloud cover decreases dramatically in the first few days following the 4 February 2012 initiation of the CGF time series, denoted here as Day 1.

Referee: This result is followed by continuation of methodology (uncertainty based on the cloud persistence count and how the CPC is recorded). I recommend division of Methodology and Results.

**Authors: We agree and have separated the Methodology and Results sections and it now reads more clearly in the revised paper.**

Referee: I agree that validation of the satellite data is only possible by comparison with measurements. The manuscript presents validation against the NOAA snow depth data provided by the dense network of meteorological stations. Such networks are not available in other countries. Can we trust that the CGF maps are valid also in those parts of the world where the network density does not allow detailed validation?

**Authors: Evaluation of the CGF maps in other countries will have to wait until the products are released and available to download through NSIDC (in the fall of 2019). In**

**areas of the world where the network of meteorological stations is not dense enough to allow validation, there are other methods to evaluate the uncertainties. These methods, discussed in the paper, include comparison with other snow maps, comparison with higher-resolution satellite data (such as with Landsat or Sentinel data), and comparison with surface reflectance maps such as from MODIS and VIIRS.**

**The MODIS SCE products have been validated and evaluated in many regions of the world; there are numerous peer reviewed articles published on this topic. However, the VIIRS SCE daily tiled product has not yet been released; only the swath product is available, so evaluation research has not yet appeared in the literature because users tend to be more comfortable using a tiled product than a swath product. In our comparisons between MODIS and VIIRS CGF products we have found very good agreement between MODIS and VIIRS SCE and CGF products thus there is the expectation that the VIIRS products will have similar accuracy to that reported for MODIS. We acknowledge, however, that the comparisons are necessarily limited because product production has not yet begun.**

Referee: I would welcome a comment on this. Is it possible to conclude that Aqua snow maps tend to have have more clouds than the Terra snow maps globally? Can it not be only the case in some areas while in others it would be vice versa?

**Authors: It is not possible to conclude that there is more cloud cover in Aqua snow maps than in the Terra maps based on our research. Cloud cover amount and location can change a little or a lot between Terra and Aqua overpasses. For example, over the Rocky Mountains in the western USA a Terra overpass at about 10:30 am local time may have a clear view of the peaks but daily convective cloud formation over the peaks may occur, and when Aqua passes over about 1:30 pm local time there could then be clouds over the peaks. Daily cloud dynamics, as your questions suggests, can vary greatly depending on location. What we are concerned about is consistent accuracy in the detection of clouds in the Terra and Aqua cloud products that are used as inputs to the swath level snow cover algorithm. Accuracy of cloud detection has an effect on the extent of snow or cloud mapped in the snow cover products.**

Referee: 4. Discussion and conclusions - I appreciate the note that CGF snow products have all uncertainties of the original products as well as additional uncertainties that are related to the age of the snow measurement (l. 445-447).

**Authors: We have clarified that statement a bit by indicating that there are additional uncertainties that are related to cloud-gap filling.**

Referee: Minor comment - text in lines 52-56 could be omitted.

**Authors: We want to put the medium-resolution Earth Science Data Record (ESDR) into context by referring to the coarse-resolution product and therefore we've decided to retain some of that paragraph. We have shortened the paragraph and focused it more into the context of the medium-resolution ESDR.**

---

## Author Response (AR1)

Dear Dr. Blöschl:

Thank you for your comments. We had previously revised the paper but we were unable to upload it when we uploaded the original response to the referees' comments. However, we have made further changes, described below, and have now been able to upload the revised manuscript.

We look forward to hearing from you regarding the revised manuscript.

Sincerely,
Dorothy Hall

**Please note that the blue bold font represents today's response, and the black bold font is copied and pasted from our earlier responses. Referees' comments are in non-bold black font.**

**We have softened and toned down the following statement: "We conclude that the MODIS Terra CGF is the more accurate MODIS snow-cover product." In the revised paper that we've uploaded, we have restated that as follows in the Abstract: According to our preliminary validation of the Terra and Aqua MODIS CGF SCE products in the western U.S. study area, we found higher accuracy of the Terra product as compared to the Aqua product. The MODIS CGF snow-cover time series may be used to extend the SCE data record from 2000, into the VIIRS era through the early 2030s and perhaps beyond.**

**In our response to the referees' comments that we had previously uploaded, we responded to comments #5 and #6 of Referee #1, and also the question about validation of snow depth posed by Referee #2. Our original responses are provided below in black font. Additional comments in response to your queries are shown in blue font.**

From Reviewer #1's review:

5) Why only temporal filter is considered for gap-filling method? During snowmelt, snow-line approach or some kind of spatial filter can be more efficient.

**Authors: There are many other useful methods of gap filling, but the method described in our paper is the method that is used to develop the new product that will be available starting this summer or fall. We are beyond the point where different methods can be considered since the new algorithm uses the CGF method, all of the programming has been completed by the MODIS Project and the products will be available soon. It is too late to change the algorithm for Collection 6.1.**

**There is no doubt that other methods of gap filling are useful and perhaps even more accurate or efficient than the method used in the NASA standard MODIS CGF product. However the method selected, as described in Hall et al. (2010) and Riggs et al. (2017b), cannot be changed because the CGF product is "in production."**

**Therefore we do not understand why Referee #1 wants us to compare gap-filling methods beyond saying that there are some very good methods out there that differ from the method that we've selected. Several years ago we selected one method based on the fact that we must produce the product very quickly after data acquisition. For example, we don't have the luxury of waiting until the clouds clear after the day in question and then looking back to fill in gaps caused by clouds. We needed an algorithm that produces a snow map within a few hours after data acquisition and we settled on the CGF algorithm described in this paper. That was decided and approved by the MODIS Project several years ago. We cannot change it now.**

6) The results show only few examples which does not allow to see clearly if the results are robust and general. More thorough analysis (longer time periods, seasonal evaluation, larger/different regions) will allow to draw much more robust findings.

**Authors: We agree with this comment, but we are unable to do a thorough and global analysis because the product is not yet being produced by the MODIS and VIIRS Projects. When processing starts, the product will be downloadable through the National Snow and Ice Data Center starting in the fall of 2019.**

**In order to develop a time series in this pre-production phase, we need to do a considerable amount of programming. We've done this by developing a Terra MODIS CGF SCE time series for the western U.S. data set for 2012. For this revised version of the paper, and in response to this and other comments, we ran a 3-month time series using VIIRS SCE maps (see Figure 9). Running a CGF time series is computationally burdensome, and therefore a comprehensive, global analysis cannot be accomplished until after the official MODIS processing begins. Even after production begins it will take many months until the complete MODIS and VIIRS time series (from 2000 to present for MODIS and from 2011 to present for VIIRS) can be processed. Complete processing is likely to occur sometime in the year 2020 for both the MODIS C6.1 and VIIRS C2 CGF SCE products.**

**Recently we found out that MODIS data processing of Collection 6.1, that includes producing the CGF snow product, will begin by early October 2019. It may take up to one year to process all of the data, globally, from 2000 – present. Until processing has been completed near the end of the year 2020, we cannot do comprehensive global validation.**

**In short, processing will not be complete in a time frame that is reasonable for providing the revisions to this paper. And it is important that this paper be published so that users of the new products will have the information contained in this paper when the products first become downloadable from NSIDC in the fall of 2019. After processing has been completed, global validation will be possible by users globally. Comprehensive global validation is not something that is possible for one person or one small group to complete.**

**Additionally, VIIRS data processing is not likely to start until later this fall. Therefore it won't be possible to even begin validating the VIIRS CGF snow products probably until early 2020. Again, many months will be required for the NASA VIIRS Project to fully process the VIIRS time series (2011 – present).**

Comment from the editor that Reviewer #2's comment about global validation was not addressed adequately:

I agree that validation of the satellite data is only possible by comparison with measurements. The manuscript presents validation against the NOAA snow depth data provided by the dense network of meteorological stations. Such networks are not available in other countries. Can we trust that the CGF maps are valid also in those parts of the world where the network density does not allow detailed validation?

**Our original response is shown below. Additional comments are shown in blue.**

**Authors: Evaluation of the CGF maps in other countries will have to wait until the products are released and available to download through NSIDC (beginning in the fall of 2019). In areas of the world where the network of meteorological stations is not dense enough to allow validation, there are other methods to evaluate the uncertainties. These methods, discussed in the paper, include comparison with other snow maps, comparison with higher-resolution satellite data (such as with Landsat or Sentinel data), and comparison with surface reflectance maps such as from MODIS and VIIRS.**

**The MODIS SCE products have been validated and evaluated in many regions of the world; there are numerous peer reviewed articles published on this topic. However, the VIIRS SCE daily tiled product has not yet been released; only the swath product is available, so evaluation research has not yet appeared in the literature because users tend to be more comfortable using a tiled product than a swath product. In our comparisons between MODIS and VIIRS CGF products we have found very good agreement between MODIS and VIIRS SCE and CGF products thus there is the expectation that the VIIRS products will have similar accuracy to that reported for MODIS. We acknowledge, however, that the comparisons are necessarily limited because product production has not yet begun.**

**It is incumbent on the user to validate the product in his/her study area. While we, the product developers, can do preliminary validation, we cannot perform global validation. For one thing we are not as knowledgeable about snow-covered areas on different continents and in different countries as are the researchers who live there.**

**Evaluation of MODIS and VIIRS Cloud-Gap Filled Snow-Cover Products for production of an Earth Science Data Record:**

Dorothy K. Hall[1,2,], George A. Riggs[2,4], Nicolo E. DiGirolamo[2,4] and Miguel O. Román[5]

[1]Earth System Science Interdisciplinary Center, University of Maryland, College Park, MD 20740, USA
[2]Cryospheric Sciences Laboratory, NASA / Goddard Space Flight Center, Greenbelt, MD 20771, USA
[3]
[4]SSAI, Lanham, MD 20706, USA

[5]Earth from Space Institute / USRA, 7178 Columbia Gateway Dr., Columbia, MD 21046, USA

*Correspondence to*: Dorothy K. Hall (dkhall1@umd.edu)

**Abstract.** MODerate resolution Imaging Spectroradiometer (MODIS) cryosphere products that have been available since the launch of the Terra MODIS in 2000 and the Aqua MODIS in 2002 include global snow-cover extent (SCE) (swath, daily and eight-day composites) at 500 m and ~5 km spatial resolution . These products are used extensively in hydrological modeling and climate studies.  augment studies of . Reprocessing of the complete snow-cover data record, from Collection 5 (C5) to Collection 6 (C6) and Collection 6.1 (C6.1), has provided improvements in the MODIS product suite. Suomi National Polar-orbiting Partnership (S-NPP) Visible Infrared Imaging Radiometer Suite (VIIRS) Collection 1 (C1) snow-cover products at 375 m spatial resolution have been available since 2011 and are currently being reprocessed for Collection 2 (C2). Both the MODIS C6.1 and the VIIRS C2 products will be available to download through the National Snow and Ice Data Center beginning in the fall of 2019, with the complete time series available in 2020. To address the need for a cloud-reduced or cloud-free daily  SCE product for both MODIS and VIIRS, a  daily cloud-gap filled (CGF) snow-cover algorithm was developed for MODIS C6.1 and VIIRS C2 processing. MOD10A1F (Terra) and MYD10A1F (Aqua) are daily, 500-m resolution  CGF  SCE map products from MODIS. VNP10A1F is the daily, 375-m resolution CGF SCE map product from VIIRS. The CGF products include quality-assurance data including cloud-persistence statistics showing the age of the  observation in each pixel. The objective of this paper is to introduce the new MODIS and VIIRS standard CGF daily SCE products and to provide preliminary evaluation of uncertainties in the gap-filling methodology so the products can be used as the basis for a moderate-resolution Earth Science Data Record (ESDR) of SCE. ~~The objective of this paper is to introduce the new MODIS and VIIRS standard CGF daily SCE products and to provide preliminary evaluation of uncertainties in the products so the products can be used as the basis for a moderate-resolution Earth Science Data Record (ESDR) of SCE. SCEprovide preliminaryion of from the CGF products, moderate resolutionSCE. Work is ongoing to evaluate and document uncertainties in the MODIS and VIIRS standard daily CGF snow-cover products. In this workT, we~~

developed MODIS and VIIRS time series of the MODIS and VIIRS CGF products and have been developed and evaluated those time series in selected study sites in the United StatesU.S. and southern Canada.  Analysis of the

MOD/MYD10A1F product accuracys for study areas in the western United States shows excellent results in terms of accuracy of snow-cover mapping.  When there are frequent clear-sky episodes, MODIS is the satellite instruments are able to capture enough clear views of the surface to produce accurate useful snow-cover information and snow maps.  Even in the extensively-cloud-covered northeastern United States during winter months, snow maps from MODIS CGF products are useful, though the snow maps are likely to miss some snow, particularly during the spring snowmelt period when snow may fall and melt within a day or two, before the clouds clear from the storm that deposited the snow.  A time-series comparison of three months of Terra MODIS and S-NPP VIIRS

CGF snow-cover maps, xx – xx, 2012, reveals xxx.  Comparisons between the Terra and Aqua CGF snow-cover maps TheObserved differences, though small, have revealed differences that are related toare largely attributed to differences in cloud masking in the two algorithms and also differences in time of day of image acquisition.

AHowever, a nearly three-month time-series comparison of Terra MODIS and S-NPP VIIRS CGF snow-cover maps for a large study area covering all or parts of 11 states in the western United States and part of southwestern Canada reveals excellent correspondence between the Terra MODIS and S-NPP VIIRS products, with a mean difference of

11,070 km² for a large (~2,487,610 km²)which is ~<0.45 percent of the study area in the western U.S. that includes all or parts of 11 states and part of southwestern Canada.  We conclud According to our preliminary validation of the

Terra and Aqua MODIS CGF SCE products in the western U.S. study area, wWe also eAdditionally, we found that t that thehigher accuracy of the Terra productMODIS  CGF is the more accurate thanas compared to  the Aqua

MODIS snow-cover product,product.  The MODIS CGF snow-cover time series  and should therefore form be the basis of an Environmental Science Data RecordESDR that willmay be used to extend the CGF SCE data record from the Terra MODIS beginning in 2000, through into the VIIRS era, at least through the early 2030s and perhaps beyond.

**1 Introduction**

Regular snow-cover mapping of the Northern Hemisphere from space began in 1966 when the National Oceanic and

Atmospheric Administration (NOAA) began started producing weekly snow maps to improve weather forecasting (Matson and Wiesnet, 1981).  A 53-year climate-data record (CDR) of Northern Hemisphere snow-cover extent (SCE), based on NOAA's snow maps is now available at the Rutgers University Global Snow Lab (Robinson et al.,

1993; Estilow et al., 2015) at a resolution of 25 km².  Since the 1960s, snow-cover mapping from space has become increasingly sophisticated.  Not only has the temporal resolution of the NOAA snow maps increased from weekly to twice-daily, but the spatial resolution has also improved over time.  Furthermore, dData from multiple satellite platforms and instruments with visible/near-infrared (VNIR) and short-wave infrared (SWIR) bands are now available to support improved snow mapping and snow/cloud discrimination as compared to the earliest satellite snow-cover maps when sparse satellite data were available.

Using tThDue to increasing global temperatures, especially in more northerly areas in the Northern Hemisphere, the

Rutgers CDR, has been used by researchers for decades, and in recent years to showhave shown that SCE has been declining and melt has been occurring earlier in the Northern Hemisphere (e.g., Déry and Brown, 2007). This shortening of the snow season has many implications such as, for example, in the western United States (Mote et al.,

2005; Stewart, 2009; Hall et al., 2015), with earlier snowmelt contributing to a longer fire season (Westerling et al.,

2006; O'Leary et al., 2018) and other environmental and societal problems. However, the coarse resolution of the

Rutgers CDR is not suitable for regional and basin-scale studies.

Meltwater from mountain snowpacks provides hydropower and water resources to drought-prone areas such as the western United States. Accurate snow measurement is needed as input to hydrological models that predict the quantity and timing of snowmelt during spring runoff. SCE can be input to models to estimate snow-water equivalent (SWE) which is the quantity of most interest to hydrologists and water management agencies.

AIncreasingly accurate predictions save money and water because reservoir management improves withas measurement accuracyknowledge of SWE improvencreases.

Moderateedium-resolution SCE maps are produced daily from multiple satellite sensors such as fromare on the

MODerate-resolution Imaging Spectroradiometer (MODIS) on both the Terra , launched in 1(1999 launch), and

Aqua (, launched in 2002 launch), and satellites, and from the Visible Infrared Imaging Radiometer Suite (VIIRS)

on the Suomi - National Polar Partnership (S-NPP) and the Joint Polar Satellite System – 1 (JPSS-1) satellites, launched in 2011 and 2017, respectively. SThese snow maps from MODIS, in particular, are used extensively by modelers and hydrologists to study regional and basin-scalelocal SCE and to develop snow-cover depletion curves for multiple hydrological and climatological applications. Algorithms utilizing data from the VIIRS and MODISse sensors provide global swath-based snow-cover maps with at spatial resolutions ranging from 375 m to 500 1 km under clear skies. Instruments on the Landsat series of satellites for which the record began in 1972, and other higher-resolution sensors, such as from the more-recent Sentinel series, provide still-higher spatial resolution data from which snow maps can be developed, though lower temporal resolution.

Cloud cover is the single most-important factor affecting the ability to map SCE accurately using visible and near infrared (VVNIR) and short-wave infrared (SWIR) sensors. Clouds often frequently create daily gaps in snow- coverSCE 
[revised manuscript text omitted]

**3 Methodology and Results**

The new standard CGF products of Collection 6.1 and C2, respectively, M*D10A1F and VNP10A1F, enable researchers to download and use cloud-free MODIS and VIIRS daily snow SCE maps along with quality-assurance (QA) data information to assess uncertainties of the gap-filling algorithm.  Reference Figure 1 here StudyHereFor the present work, we focus on a large (2,487,610 km²) study area covering all or parts of 11 states in the western

U.S. and part of southern Canada (Figure 1).  Examples of the daily Terra MODIS standard and CGF and the daily

S-NPP VIIRS standard and CGF cloud-free map products for thise western U.S. study area (Fig. 1) may beare seen in Fig. 2.  Note someThere are some differences in cloud cover between the Terra MODIS (top left) and S-NPP

VIIRS (top right) standard snow maps.  The MOD10A1F scenesnow map is 65.8 percent (1,637,066 km²) cloud- covered, vs 60.6 percent (1,506,924 km²) in the VNP10A1F snow map.  The difference in cloud coverage is largely due to the differences in the cloud masking of MODIS and VIIRS SCE maps, as described earlier.  However, difference in the locations of clouds is also a contributing factor because the Terra MODIS and S-NPP VIIRS

images were acquired at different times on the same day, and clouds move.  There may also be changes in the location of snow cover within a day (due to melting of shallow snow, for example).  Even given these small differences in the standard products that include clouds, the CGF snow maps shown in the bottom row of Fig. 2 are very similar, with 15.2 percent (378,634 km2) snow cover on the MOD10A1F snow map and 16.6 percent (413,794

km2) snow cover on the VNP10A1F snow map.  Thus the VIIRS maps shows fewer clouds and more snow than does the Terra MODIS map in this example.

in area.Examples of the dThe daily Terra MODIS CGF S-NPP SCE product is similar to MOD10A1 product but is cloud-free for the western U.S. study area (Fig. 1),  as seen in Fig. 2.  There are some differences in cloud cover between the Terra MODIS (top left) and S-NPP VIIRS (top right) snow maps.  .The percentage of clouds in the

MOD10A1F scene is 65.8 percent (1,637,066 km2), vs 60.6 percent (1,506,924 km2) in the VNP10A1F snow map.

Th difference in cloud coverage isis is largely due to the differences in the cloud masking of MODIS and VIIRS, described earlier.  It isHowever dare also a factor because the images were acquired at different times on the same day, and clouds moveTperhaps even some changes in snow cover.  The CGF snow maps shown in the bottom row of Fig. 2 are very similar, with 15.2 percent (378,634 km2) snow cover on the MOD10A1F snow map and 16.6

percent (413,794 km2) snow cover on the VNP10A1F snow map, with VIIRS modis/viirsmapping more snow than

MODIS.

the accuracy of the snow observation depends in part on the age of the observation, i.e., number of days since last cloud-free observation, thus information on cloud persistence is included with each product.  The accuracy of the observation at the pixel level depends on the cloud masking of the swath product, M*D10_L2, for MODIS and

VNP10_L2 for VIIRS.  The MODIS and VIIRS snow-cover swath products are gridded and mapped into the daily tiled products that are input to M*D10A1F and VNP10A1F CGF algorithms.

[Figure]

**Figure 1:**

[Figure]

[Figure]

**Figure 2:**  Examples of MODIS and

VIIRS standard and cloud-gap filled (CGF) snow maps on 14 April 2012 for a study area in the western United

States/southwestern Canada (see **Fig. 1**). **Top**  **left**: MODIS MOD10A1 C6.1 snow map showing extensive cloud cover on

14  April 2012. **Top right**: VIIRS VNP10A1 C1 snow map also showing extensive cloud cover on 14 April 2012.

**Bottom left**: MOD10A1F C6.1 CGF map corresponding to the MOD10A1 snow map in the top row, also for 14  April 2012. **Bottom right**: VNP10A1F CGF map corresponding to the VNP10A1 snow map in the top row, also for 14 April 2012. In all of the snow maps, Nnon-snow-covered land is green. Regions of interest  containing the

Sierra Nevada Mountains in California and Nevada (109,575 km$^2$), and the Wind River Range in Wyoming (22,171 km$^2$), are outlined in red on the MODIS snow maps. The following MODIS tiles were used to develop the MODIS composites: h08v04, h09v04, h10v04, h08v05, h09v05, h10v05. Each VIIRS swath that included coverage of this study area was composited to create a daily map, then the daily maps were used to create the VNP10A1F snow map for 14 April 2012.

**REVISE this FIGURE.  Instead of showing the 15 April 2012 MODIS image pair, replace that with 14 April VIIRS snow**

**maps.**

Though cloud-gap filling provides a cloud-free snow map every day, the accuracy of the snow observation depends in part on the age of the observation, i.e., number of days since last cloud-free observation, thus information on cloud persistence is included with each product. The accuracy of the observation at the pixel level depends on the snow-cover algorithm that includes cloud masking of the swath product, M*D10_L2, for MODIS and VNP10_L2 for VIIRS. The MODIS and VIIRS snow-cover swath products are gridded and mapped into the daily tiled products which that are input to M*D10A1F and VNP10A1F CGF algorithms.

The accuracy of athe snow observation is dependent on many factors. In this work, we focus on the uncertainties of the gap-filling method; we do not address the inherent accuracy of the snow maps because that has been documented elsewhere by many previous studies, at least for the MODIS SCE products. The accuracy ofUncertainties in the CGF maps that relate to the gap-filling methodology, shown in Fig. 2 depends in part on the age of the observation, i.e., number of days since last cloud-free observation. To address this, information on cloud persistence for each pixel is included with each product. The accuracy of the observation at the pixel level also depends on the Celoud masking of the swath product, M*D10_L2, for MODIS and VNP100_L2 for VIIRS, represents an additional uncertainty in the both products and contributes to differences between the snow-mapping results. The MODIS and VIIRS snow-cover swath products are gridded and mapped into the daily tiled products that are input to M*D10A1F and VNP10A1F CGF algorithms (Riggs et al., 2017a).

IFor MODIS, inputs to the MODIS CGF algorithms are the current day M*D10A1 and the previous day M*D10A1F products. The CGF daily snow map is created by replacing cloud observations in the current day M*D10A1 with the most-recent previous cloud-free observation from the M*D10A1F (Hall et al., 2010; Riggs et al., 2018). The algorithm tracks the number of days since the last cloud-free observation by incrementing the count of consecutive days of cloud cover for a pixel. This is stored in the cloud-persistence count (CPC) data array. If the current day observation is 'cloud' then the cloud count is one and is added to the CPC count from the previous day's M*D10A1F and written to the current day's M*D10A1F algorithm. If the current day observation is 'not cloud,' then the CPC is reset to zero in the current day's M*D10A1F CPC. If the CPC is 0, that means that the snow-cover observation is from the current day. If the CPC for the current day is ≥1, that represents the count of days since the last 'non-cloud' observation. On the day that the CGF mapping algorithm is initialized for a time series., for example, 1 xx February 2012, the CGF snow-cover map is identical to the MODIS daily snow-cover map (M*D10A1) and the cloud-persistence count (CPC) map will show zeros for non-cloud observations and ones for cloud observations (Riggs et al., 2018). As the time series progresses, a nearly-cloud-free snow map is produced on about Day 5-8 in theis example, shown in Fig. 3. on whichwhen the percent clouds cover is only 3.88.0 percent of the snow map (Fig. 3), though it takes 24 days to achieve a completely cloud-free map in this example (not shown). The same method is used to develop the VNP10A1F CGF snow-map products. For the same initialization of the time series, beginning on 4 February 2012, a nearly-cloud-free snow map is produced on Day 5x when the clouds cover is only xxx6.7 percent of the map, and it takes xxx days to achieve a completely cloud-free VNP10A1F CGF snow map (Fig. 3).

[Figure]

**Figure 3:**
initiation

[revised manuscript text omitted]

For the study area in the western U.S. shown in Figure 1, A ~3-month (14 Februaryxx – 30 April xx, 2012) time series of Terra MODIS and S-NPP VIIRS SCE map products (Fig. 9) wasere developed, processed and evaluated for the study area in the western U.S. shown in Fig. 1.  Note in Figure 9 theThe difference in SCE between the MODIS and VIIRS snow maps for each day of thethe time series is shown in the graph.  Overall, the snow maps agree very well though t.he mean difference shows that In general, the Terra MODIS snow maps show more/less snow as compared to the VIIRS snow maps, with a mean daily difference of -11,070 sq km$^2$..,which represents only 0.45 percent of the study area which is only ~0.45 percent of the study area.. Overall, the snow maps agree very well.  Reasonss for disagreement between MODIS and VIIRS on a given day daily basis are that the Terra MODIS images are acquired at a different time of the day (10:30 A.M. equatorial crossing time) as compared to the S-NPP VIIRS images (1:30 P.M. equatorial crossing time); cloud-cover differences on the original snow maps (before gap filling) can also explain some of the difference in amounextent of snow mapped.  This is largelys because of differences in cloud masking between the MODIS and VIIRS SCE products as described earlier. in Section xxand as illustrated in the example shown in Fig. 2..

Further analysis has confirmed that the differences in SCE between the two snow maps are largely due to differences in cloud masking.

[revised manuscript text omitted]

respectively.  The objective of this work the NASA MODIS and VIIRS algorithms products is to produce a daily,
cloud-free snow-cover products along with appropriate QA information.  These products will enable that can SCE
can be used as the basis for that an Earth Science Data Record (ESDR) of snow cover toean be produced at moderate
spatial resolution for hydrological and climatological applications.  Cloud-gap filled snow-cover products from
MODIS and VIIRS have all of the uncertainties of the original products, that contain clouds, as well as additional
uncertainties that are related to cloud-the age of the snow measurementgap -filling, such as the age of the snow
observation method.  When using the MODIS and VIIRS CGF products, a user can specify how far back in time
they want to look, using the Cloud-Persistence Count (CPC) which tells the age of the snow measurement in each
pixel; the CPC, and is available as part of the product QA metadata for both the MODIS and VIIRS CGF snow-
cover products.  Uncertainty relating to cloud-gap filling is greater in areas with frequent and persistent cloud cover
during the snow season such as in the northeastern U.S., or WRR vs. areas such as the Sierra Nevada Mountains
where gaps in clouds occur more frequently during the snow season.
It is difficult to validate the MODIS and VIIRS CGF (and other) snow maps.  Absolute validation can only be
accomplished using  NOAA daily snow depth station data when available.  However, pwWe can also evaluate the
product accuracy can also be *evaluated* by comparing the CGF with MODISproducts 
[revised manuscript text omitted]

---

## Author Response (AR2)

Dear Dr. Blöschl:

Thank you very much for handling our paper.  Following the review process, our paper has improved greatly and we are appreciative.

We have improved all of the figures and uploaded them separately.  The quality of the individually-uploaded figures is better than is shown in the final PDF version that is also uploaded.  We made all of the suggested corrections to the figures.

We also corrected the section numbering.

We look forward to hearing about the final status of the paper.

Sincerely,
Dorothy Hall